# Key challenges facing the application of the conductivity mass balance method: a case study of the Mississippi River Basin

Hang Lyu[1,2], Chenxi Xia[1,2], Jinghan Zhang[1,2], Bo Li[1,2]

[1]Key Laboratory of Groundwater Resources and Environment (Jilin University), Ministry of Education, Changchun 130026, China.

[2]Jilin Provincial Key Laboratory of Water Resources and Environment, Jilin University, Changchun 130026, China.

*Correspondence to: Hang Lyu(lvhangmail@163.com)*

**Abstract:** The conductivity mass balance (CMB) method has a long history of application to baseflow separation studies. The CMB method uses site-specific and widely available discharge and specific conductance data. However, certain aspects of the method remain unstandardized, including the determination of the applicability of this method for a specific area, minimum data requirements for baseflow separation and the most accurate parameter calculation method. This study collected and analyzed stream discharge and water conductivity data for over 200 stream sites at large spatial (2.77 km² to 2,915,834 km² watersheds) and temporal (up to 56 years) scales in the Mississippi River Basin. The suitability criteria and key factors influencing the applicability of the CMB method were identified based on an analysis of the spatial distribution of the inverse correlation coefficient between stream discharge and conductivity and the rationality of baseflow separation results. Sensitivity analysis, uncertainty assessment and T-test were used to identify the parameter in the method was most sensitive to, and the uncertainties of baseflow separation results obtained from different parameter determination methods and various sampling durations were compared. The results indicated that the inverse correlation coefficient between discharge and conductivity can be used to quantitatively determine the applicability of the CMB method, while the CMB method is more applicable in tributaries, headwater reaches, high altitudes and regions with little influence from anthropogenic activities. A minimum of six-month discharge and conductivity data was found to provide reliable parameters for the CMB method with acceptable errors, and it is recommended that the parameters $SC_{RO}$ and $SC_{BF}$ be determined by the 99th percentile and dynamic 99th percentile methods, respectively. The results of this study can provide an important basis for the standardized treatment of key problems in the application of the CMB.

## 1. Introduction

Baseflow is the ground water contribution to total stream flow (Hewlett and Hibbert, 1967), which plays a critical role in sustaining streamflow during dry periods (Rosenberry and Winter, 1997). Quantitative estimates of stream baseflow can be used to determine baseflow response to environmental conditions, thereby improving understanding of the water budget of a watershed and facilitating the estimation of groundwater discharge and recharge (Tan et al., 2009; Dhakal et al., 2012; Ran et

al., 2012).

Given the importance of baseflow, many methods have been proposed for baseflow separation. Although these methods can be categorized according to various conditions (Stewart et al., 2007; Zhang et al., 2012; Miller et al., 2014; Lott and Stewart, 2016), they can generally be divided into two groups, namely non-tracer-based and tracer-based separation methods (Li et al., 2014). Non-tracer methods mainly include graphical and low-pass filter methods which only require stream discharge data (Nathan and McMahon, 1990; Eckhardt, 2008). Given the wide availability of stream discharge records, these approaches can readily be applied to a large number of sites (Miller et al., 2014). However, since these methods are typically applied without reference to any hydrological basin variables, the objective assessment of their accuracy remains a challenge (Nathan and McMahon, 1990; Arnold and Allen, 1999; Arnold et al., 2000; Furey and Gupta, 2001; Huyck et al., 2005; Eckhardt, 2008). In contrast, tracer-based baseflow separation methods adhere to the principle of mass balance (MB). Tracers such as stable isotopes, major ions and specific conductance (SC) have been used to quantify surface runoff and groundwater discharge to streamflow (Miller et al., 2014). The advantage of these methods relates to their use of site-specific variables, such as concentrations of chemical constituents, which are a function of actual physical processes and flow paths in the basin responsible for generation of different flow components. Therefore, chemical mass balance estimates of baseflow are often considered to be more reliable than those from graphical hydrograph separation estimates (Stewart et al., 2007). The principal disadvantage of mass-balance methods relates to their requirements for both observed discharge and chemical concentration data, which are not widely available, especially over a long period. This makes the application of the MB method in large basins impractical over a long period. For example, while stable isotopes are generally considered to be the most accurate chemical tracers for hydrograph separation (Kendall and McDonnell, 2012), the analytical costs associated with these constituents often limit their use in large studies (Miller et al., 2014).

In an analysis of hydrograph separation conducted using different geochemical tracers, Caissie et al. (1996) demonstrated that specific conductance (SC) was the most effective single variable for quantifying the runoff and groundwater components of total streamflow since SC is a natural environmental tracer that can be inexpensively measured concurrently with stream flow. (Kunkle, 1965; Matsubayashi et al., 1993; Arnold et al., 1995; Caissie et al., 1996; Cey et al., 1998; Heppell and Chapman, 2006; Stewart et al., 2007; Pellerin et al., 2008).

The CMB method converts specific conductance to a baseflow value using a two-component mass balance calculation (Pinder and Jones, 1969; Nakamura, 1971; Stewart et al., 2007):

$$Q_{BF} = Q\left[\frac{SC - SC_{RO}}{SC_{BF} - SC_{RO}}\right] \qquad (1)$$

In Eq. (1), $Q$ is the measured streamflow discharge ($L^3T^{-1}$), SC is the measured specific conductance ($lS\ cm^{-1}$) of streamflow, $SC_{RO}$ is the specific conductance of the runoff end-member, and $SC_{BF}$ is the specific conductance of the baseflow end-member.

Certain questions need to be addressed before the CMB method can be considered for separating baseflow in a watershed. These include whether the CMB method is applicable to a watershed, how to more accurately determine the key parameters $SC_{RO}$ and $SC_{BF}$ when a long series of monitoring data are available, and the length of the monitoring period required to ensure

the accuracy of the results when adopting a CMB method for a new conductivity monitoring network. These questions have been partially answered by past studies. Miller et al. (2014) concluded that the CMB method was successful in quantifying baseflow in a variety of stream ecosystems, including snowmelt-dominated watersheds (Covino and McGlynn, 2007), urban watersheds (Pellerin et al., 2008) and a range of other settings (Stewart et al., 2007; Sanford et al., 2011; Lott and Stewart, 2016). However, most chemical hydrograph separation studies have been conducted in small watersheds and for short durations (Miller et al., 2014). In addition, the CMB method is often not appropriate for application to systems in which there is not a consistent inverse correlation between discharge and SC, particularly for sites heavily influenced by anthropogenic activities. However, there appears to be no further systematic summary of characteristics of watershed systems that indicate the suitability of the CMB method. Questions therefore remain of how to determine whether the CMB method is appropriate for application to a particular watershed, and which factors have the greatest impact on the outcome of the application of the CMB method. Further uncertainties in the CMB method relate to appropriate methods for determining the parameters of the method. Stewart et al. (2007) determined through a field test that the maximum and minimum conductivity can be used to replace $SC_{BF}$ and $SC_{RO}$, respectively. Miller et al. (2014) found that the maximum conductivity of streamflow may exceed the real $SC_{BF}$; therefore, they suggested the use of the 99th percentile of conductivity of each year as $SC_{BF}$ to avoid the impact of high $SC_{BF}$ estimates on the separation results and assumed that baseflow conductivity varies linearly between years. However, questions remain in relation to which parameter determination method is more reasonable and accurate for calculation of baseflow. In a study of the shortest monitoring period of the CMB method, Li et al. (2014) evaluated data requirements and potential bias in the estimated baseflow index (BFI) using conductivity data for different seasons and/or resampled data segments at various sampling durations, and found that a minimum of six months discharge and conductivity data are required to obtain reliable parameters with acceptable errors. However, their study conceded that further studies of watersheds at large temporal and spatial scales are need to verify the conclusions.

The present study conducted a comprehensive qualitative and quantitative analysis of data from more than 200 hydrological sites widely distributed in the Mississippi River Basin, United States of America. Based on the results of statistical analysis, the present study had the following objectives: (1) Determine the criteria and main factors influencing the applicability of CMB method; (2) Identify the best method for determining the parameters of the CMB method; (3) Determine data requirements for the CMB method. The conclusions of the present study can help to determine whether the CMB method is applicable to a particular river reach and can provide a reference standard for use of the method.

## 2. Methods

### 2.1 Data sources and site description

The Mississippi River Basin is located on the western side of the continental divide. The basin encompasses five states and has a drainage area of 320,000 km$^2$. A total of 201 sites were selected in watersheds of the Missouri, Illinois, Minnesota, Iowa, Ohio, Arkansas, Red, White and Des Moines rivers to represent the variability of sub-basin areas and physiographic and

climatic regions, with the areas of sub-basins ranging from 2.77 km² to 2,915,834 km² (Fig. 1). Each selected site had at least two years of continuous discharge data paired with specific conductance data. All discharge and specific conductance data used in the present study were mean daily values retrieved from the United States Geological Survey's (USGS) National Water Information System (NWIS) website (http://waterdata.usgs.gov/nwis).

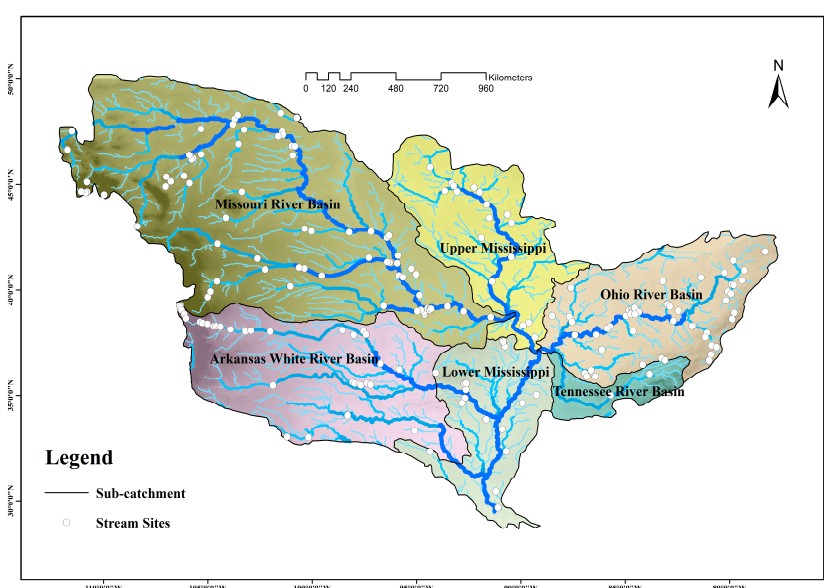

**Figure 1. Map showing the Mississippi River Basin and the locations of the 201 stream gauging sites included in the present study.**

## 2.2 Determination of the applicability of the CMB method and the identification of the major factors influencing the applicability of the CMB method

The CMB method assumes that the two main recharge sources in any particular river section, streamflow runoff and baseflow, have relatively stable conductivity values (Stewart et al., 2007; Lott and Stewart, 2012). Under natural conditions, streamflow conductivity reaches a maximum value under the dry season minimum discharge, indicating the dominant contribution of baseflow to streamflow (Miller et al., 2014). In contrast, streamflow conductivity will decrease during the high flow period when the contribution of direct runoff through rainfall or snowmelt to discharge increases. This relationship

between stream conductivity and the discharge persists through intermediate state stream flows, with an inverse power function between streamflow discharge and conductivity identified (Miller et al., 2014). Conditions under which the above general relationship does not apply indicate the influence of other external factors on the river which the CMB method would be unable to represent. Therefore, during the process of baseflow separation, the applicability of the CMB method to a particular river section can be determined by identifying the relationship between stream discharge and conductivity.

In the present study, to identify the applicability of the CMB method to the 201 different site locations in the Mississippi River Basin, the relationships between conductivity and streamflow discharge at the sites were quantitatively evaluated by

correlation analysis. Stream sites were grouped into four categories according to the strength of the relationship, as indicated by the inverse correlation coefficient (r): (1) high degree of inverse correlation (r ≤ −0.8); (2) medium degree of inverse correlation (−0.8 < r ≤ −0.5); (3) low degree of inverse correlation (−0.5 < r ≤ −0.3); (4) no inverse correlation (r > −0.3). The present study analyzed the spatial distribution of stream site correlation coefficients in the basin combined with statistical data on topography, stream discharge and anthropogenic activities. The influences of these factors on the inverse correlation were studied, following which the key factors affecting the applicability of the CMB method to sub-basins of different spatial scales were identified. Thus, a set of judgement criteria for the applicability of the CMB method for baseflow separation to a certain area was established.

### 2.3 Determination of the $SC_{BF}$ and $SC_{RO}$

As according to the CMB equation [Eq. (1)], the key parameters that are needed to calculate the baseflow index of total flow are the conductivities of baseflow ($SC_{BF}$) and surface runoff ($SC_{RO}$). It is generally believed that runoff dominates streamflow during the extreme high-flow and the minimum stream conductivity periods of each year, during which stream conductivity is assigned as $SC_{RO}$. In contrast, stream conductivity during extreme low-flow and maximum stream conductivity periods of each year is assigned as $SC_{BF}$, during which baseflow dominates streamflow (Stewart et al., 2007; Lott and Stewart, 2012).

Several approaches are currently used to determine $SC_{BF}$: (1) directly assigning the maximum stream conductivity of the stream monitoring record as $SC_{BF}$ (Stewart et al., 2007); (2) assigning the 99[th] percentile (ordered by increasing conductivity) of the stream conductivity monitoring record to avoid the impacts of extremely high $SC_{BF}$ estimates that may arise when river conductivity has been affected by factors such as evaporation, irrigation, mining activity and the use of salts as road de-icing agents on the separation results; (3) identifying yearly dynamic maximum or 99[th] percentile conductivity measurements within a monitoring record as $SC_{BF}$ (Miller et al., 2014).

Since Stewart et al. (2007) has pointed out that longer conductivity records are more likely to contain low conductivity values associated with high-discharge, the present study used the minimum or 99[th] percentile (ordered by decreasing conductivity) method to estimate $SC_{RO}$.

The sensitivities of BFI to $SC_{BF}$ and $SC_{RO}$ expressed as an index, i.e. S(BFI/ $SC_{BF}$) and S(BFI/ $SC_{RO}$), respectively, and the uncertainties of $SC_{BF}$, $SC_{RO}$ and BFI, which can be expressed as $W_{SCBF}$, $W_{SCRO}$ and $W_{BFI}$, respectively, were calculated using the monitoring data of 26 stream sites with long-term records of stream discharge and conductivity for at least 5 years. The present study then proposed an optimal method of determining $SC_{BF}$ and $SC_{RO}$ according to an analysis of different methods for calculating baseflow hydrographs.

### 2.4 Data requirements for $SC_{BF}$ and $SC_{RO}$

Monitoring data of 26 stream sites with long-term records of stream discharge and water conductivity were analyzed to study the influence of different monitoring durations on the accuracy of parameter determination and baseflow separation

results. Among the 26 sites, 5 had monitoring periods longer than 14 years, whereas the remainder had monitoring periods longer than 5 years. Continuous sampling periods within the 5 longer stream monitoring records included 3, 6, 9, 12, 15, 18, 21 and 24 months, whereas those in the remaining stream monitoring records included 3, 6, 9 and 12 months. To reduce the sampling error caused by the small number of samples, overlapping of monitoring data was allowed when sampling. In addition, each segment for a specific sampling duration was randomly chosen due to the variability in water quality measurements (Li et al., 2014). $SC_{BF}$, $SC_{RO}$ and BFI were calculated for each segment, following which it was determined whether the BFI of all segments for the specific sampling durations followed normal distributions. On the premise of following a normal distribution, the BFI values obtained using 3, 6, 9, 12, 15, 18 and 21 months of conductivity measurements were compared with the BFI values obtained with 24 months data for the 5 sites with longer records. For the remaining sites, the BFI values obtained with 3, 6 and 9 months conductivity measurements were compared with the BFI values obtained with 12 months of data. A student's T test at a statistical significance level of 0.05 was used to examine the differences between BFI determined from data of each sampling duration and those from the 24 months or 12 months of data. No significant difference in BFI values estimated with a shorter duration of conductivity records with those obtained with 24 months or 12 months of data ($P > 0.05$) indicated that the shorter time duration for conductivity measurement was acceptable.

**2.5 Quantitative estimates of the sensitivity and uncertainty in baseflow**

As mentioned above, the sensitivities of BFI measurement to $SC_{BF}$ and $SC_{RO}$ were calculated and the uncertainties of CMB results obtained using different parameter determination methods and monitoring durations were evaluated to identify the most accurate parameter calculation method and the shortest appropriate monitoring period.

The dimensionless sensitivity index of BFI (output) with $SC_{BF}$ (uncertain input) and $SC_{RO}$, S(BFI| $SC_{BF}$) and S(BFI| $SC_{RO}$), reflecting the proportional relationship between the relative error in BFI and the relative error in parameters, were calculated using the following equations (Yang et al., 2019):

$$\text{S(BFI}|SC_{BF}) = \frac{SC_{BF}(ySC_{RO} - \sum_{k=1}^{n} y_k SC_k)}{yBFI(SC_{BF} - SC_{RO})^2} \tag{2}$$

$$\text{S(BFI}|SC_{RO}) = \frac{SC_{RO}(\sum_{k=1}^{n} y_k SC_k - ySC_{BF})}{yBFI(SC_{BF} - SC_{RO})^2} \tag{3}$$

In Eq. (2) and Eq (3), $y$ is streamflow ($L^3T^{-1}$) and $k$ is the time step.

There is uncertainty associated with the estimation of true means from finite samples, which is regarded as a type of error in statistical inference (Lo, 2005). This uncertainty in the CMB method was estimated based on the uncertainties in $SC_{BF}$, $SC_{RO}$, and $SC_k$. Under the approach used in the present study, the errors in the input variables are propagated to output variables following the uncertainty transfer equation derived from Genereux and Hooper (1998):

$$W_{fbf} = \sqrt{\left(\frac{f_{bf}}{SC_{BF} - SC_{RO}} W_{SCBF}\right)^2 + \left(\frac{1 - f_{bf}}{SC_{BF} - SC_{RO}} W_{SCRO}\right)^2 + \left(\frac{1}{SC_{BF} - SC_{RO}} W_{SCK}\right)^2} \tag{4}$$

In Eq. (4), $f_{bf}$ is the ratio of baseflow to streamflow in a single calculation process, $W_{fbf}$ is the uncertainty in $f_{bf}$ at the 95% confidence interval, $W_{SCBF}$ is the standard deviation of $SC_{BF}$ multiplied by the t-value (α = 0.05; two-tail) from the Student's distribution, $W_{SCRO}$ is the standard deviation of the lowest 1% of measured SC concentrations multiplied by the t-value (α = 0.05; two-tail), and $W_{SCK}$ is the analytical error in the SC measurement multiplied by the t-value (α = 0.05; two-tail). The average uncertainty in multiple calculation processes is then used to estimate the uncertainty in the baseflow index (BFI, long-term ratio of baseflow to total streamflow), which can be expressed as $W_{BFI-Genereux}$ (Genereux and Hooper, 1998; Miller et al., 2014).

On the other hand, Yang et al. (2019) found that random measurement errors in $y_k$ or $SC_k$ for time series exceeding 365 days will cancel each other out, allowing the influence on BFI to be ignored. An additional uncertainty estimation method of BFI can then be derived on the basis of the sensitivity analysis (Yang et al., 2019):

$$W_{BFI-Yang} = \sqrt{(S(BFI|SC_{BF})\frac{BFI}{SC_{BF}}W_{SCBF})^2 + (S(BFI|SC_{RO})\frac{BFI}{SC_{RO}}W_{SCRO})^2} \tag{5}$$

In Eq. (5), $W_{SCBF}$ and $W_{SCRO}$ represent the same type of uncertainty values for $SC_{BF}$ and $SC_{RO}$, respectively, as described above (Yang et al., 2019).

Given that the determination of the parameters involves sensitivity analysis, and that the sampling period of the shortest time series might not exceed 1 year, both the uncertainty estimation methods of BFI proposed by Yang et al. (2019) and Genereux and Hooper (1998) were used to determine the parameters and the shortest time series in the present study.

## 3. Results

### 3.1 Assessment of sub-basin criteria for suitability of the CMB method

The analysis of the 201 stations across the major Mississippi River Basin showed a high variation in response of conductivity to stream discharge. Most sites (157) showed an inverse correlation between streamflow discharge and conductivity, with the number of sites with the high, medium, and low inverse correlations being 47, 72 and 38, respectively. The goodness of fit ($R^2$) of each site identified by regression analysis ranged from 0.00002 to 0.9655 (Fig. 2).

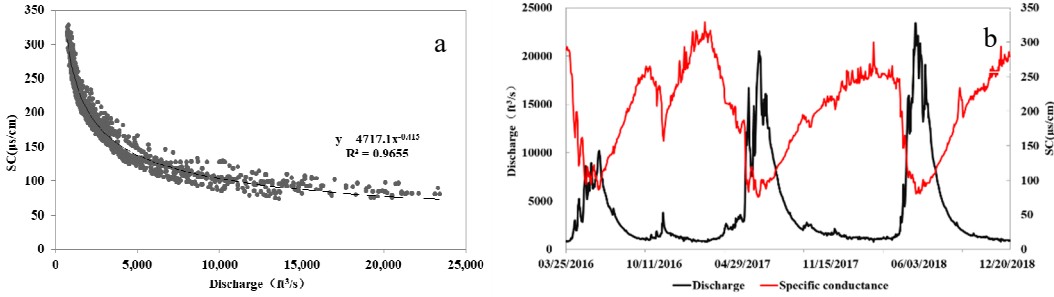

Yellowstone River at Corwin Springs MT,Site No: 06191500

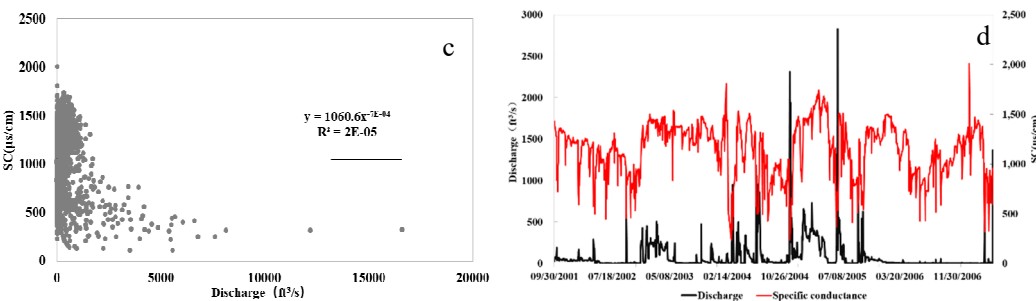

North Canadian River below Lake Overholser near OKC, OK, Site No: 07241000

**Figure 2. Inverse correlation between stream discharge and conductivity (a, c) and their temporal variation (b, d)**

An analysis of the spatial distribution of inverse correlations between stream discharge and conductivity in the basin showed that the correlations were related to various factors including topography, altitude, stream discharge and location. In general, most stations located in stream headwater areas with a steep terrain and high elevation showed inverse correlations between flow and conductivity, with 18/19 of the sites with an elevation above 1,500 m showing an $r \leq -0.5$. Fewer sites (101/182) falling within middle and lower reaches with a lower topography showed an $r \leq -0.5$ (Fig. 3). These results showed that sites with an inverse correlation between conductivity and streamflow were more likely to be located on tributaries than on mainstems. The proportions of sites in which the correlation coefficient $r \leq -0.5$ for mainstems and tributaries for the Missouri River Basin, Upper Mississippi River Basin, Lower Mississippi River Basin, and Ohio River Basin were 36.4% (4/11) and 51.6% (33/64), 50% (3/6) and 54.5% (6/11), 0% and 77.8% (14/18), and 50% (5/10) and 70.5% (31/44), respectively. On the other hand, the quantitative relationship between streamflow discharge and the correlation coefficient was not significant, and there were significant differences among the stream discharges of sub basins.

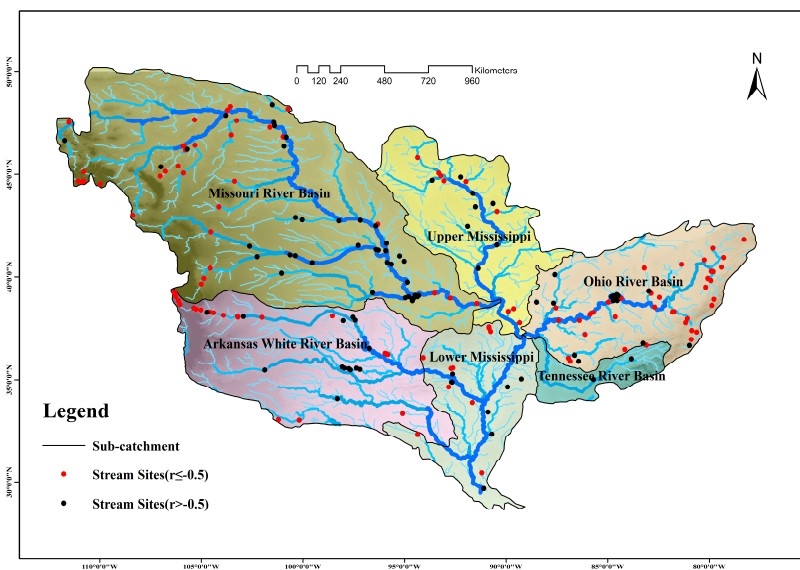

**Figure 3. Spatial distribution of data analysis points within the Mississippi River Basin according to the correlation**

### 3.2 Comparison of different SC$_{BF}$ and SC$_{RO}$ determination methods

The sensitivity analysis results (Table 1) showed that the sensitivity indices of BFI for SC$_{BF}$ and SC$_{RO}$ were all negative, indicating negative correlations between BFI and SC$_{BF}$ (SC$_{RO}$). The absolute value of the sensitivity index for SC$_{BF}$ was generally greater than that for SC$_{RO}$, indicating that BFI was affected by SC$_{BF}$ to a greater degree. Taking the site 07097000 as an example, uncertainty of 10% for both SC$_{BF}$ and SC$_{RO}$ resulted in the contribution of SC$_{BF}$ to the uncertainty in BFI being −1.34 times 10% (−13.4%), whereas that of SC$_{RO}$ was −0.56 times 10% (−5.6%). Therefore, it is clear that more attention

should be focused on SC$_{BF}$ to reduce uncertainty in BFI.

    On this basis, the uncertainty values W$_{SCBF}$ and W$_{BFI-Yang}$ obtained from different determinations of SC$_{BF}$ were compared, with the yearly dynamic maximum and yearly dynamic 99$^{th}$ percentile determination methods mainly considered. This approach was adopted as anthropogenic activities over long periods of time or year-to-year changes in the water table level may result in temporal changes in SC$_{BF}$ (Miller et al., 2014). Therefore, by adopting yearly dynamic maximum and 99$^{th}$

percentile values, the effects of temporal fluctuations in SC$_{BF}$ can be avoided. The results showed that nearly all the uncertainty values W$_{SCBF}$ and W$_{BFI-Yang}$ obtained from using the yearly dynamic 99$^{th}$ percentile were less than the corresponding values obtained from yearly dynamic maximum values. In addition, the values of W$_{SCRO}$ were much less than those of W$_{SCBF}$, which can be explained by considering that W$_{SCRO}$ is the standard deviation of the lowest 1% of measured SC concentrations multiplied by the t-value ($\alpha = 0.05$; two-tail). This excluded the possibility of calculating various standard deviations; therefore,

various W$_{SCRO}$ have not been compared in the present study.

**Table 1. A comparison of results for different methods used to obtain parameters for baseflow separation methods**

| Site number | Drainage area (square miles) | Elevation (m) | Slope (°) | S(BFI\|$SC_{BF}$) | | S(BFI\|$SC_{RO}$) | | WSC$_{BF}$ | | WSC$_{RO}$ | W$_{BFI-Yang}$ | |
|---|---|---|---|---|---|---|---|---|---|---|---|---|
| | | | | 1 | 2 | 1 | 2 | 1 | 2 | | 1 | 2 |
| 07097000 | 4,024 | 1,537 | 1.27 | −1.28 | −1.34 | −0.59 | −0.56 | 159.76 | 108.92 | 16.71 | 0.11 | 0.10 |
| 07119700 | 10,901 | 1,302 | 1.23 | −1.29 | −1.47 | −0.83 | −0.85 | 1,291.34 | 285.55 | 50.87 | 0.28 | 0.09 |
| 07086000 | 427 | 2,727 | 3.02 | −1.53 | −1.56 | −1.50 | −1.47 | 41.18 | 32.48 | 3.93 | 0.08 | 0.07 |
| 06711565 | 3,391 | 1,606 | 0.38 | −1.04 | −1.11 | −0.90 | −0.90 | 1,007.86 | 770.69 | 30.02 | 0.11 | 0.12 |
| 06089000 | 1,774 | 1,017 | 1.11 | −0.91 | −1.15 | −0.62 | −0.64 | 1,119.02 | 560.66 | 31.09 | 0.23 | 0.22 |
| 03007800 | 248 | 449 | 0.59 | −1.45 | −1.72 | −2.82 | −3.06 | 47.57 | 24.62 | 5.99 | 0.09 | 0.08 |
| 03036000 | 344 | 320 | 3.17 | −2.10 | −2.21 | −2.01 | −2.02 | 163.78 | 157.80 | 23.49 | 0.15 | 0.16 |
| 03044000 | 1,358 | 270 | 10.68 | −1.18 | −1.22 | −0.78 | −0.76 | 288.93 | 132.95 | 27.93 | 0.09 | 0.06 |
| 03067510 | 60 | 1,085 | 0.65 | −1.25 | −1.46 | −1.69 | −1.82 | 42.81 | 17.97 | 4.58 | 0.16 | 0.11 |
| 03072655 | 4,440 | 242 | 9.51 | −1.31 | −1.38 | −1.47 | −1.46 | 114.27 | 69.93 | 12.12 | 0.06 | 0.05 |
| 03073000 | 180 | 262 | 1.40 | −1.34 | −1.37 | −1.50 | −1.49 | 1,900.59 | 1,920.89 | 32.96 | 0.10 | 0.12 |
| 03106000 | 356 | 264 | 4.60 | −1.31 | −1.32 | −1.21 | −1.15 | 439.54 | 370.16 | 30.99 | 0.11 | 0.11 |
| 03199700 | 837 | 183 | 7.10 | −1.61 | −1.57 | −1.51 | −1.44 | 385.18 | 366.02 | 16.69 | 0.11 | 0.12 |
| 03201980 | 100 | 194 | 0.83 | −1.27 | −1.42 | −1.32 | −1.35 | 374.47 | 270.71 | 42.43 | 0.09 | 0.09 |
| 03238745 | 39 | 170 | 2.22 | −0.62 | −0.54 | −0.69 | −0.59 | 2,075.52 | 1,959.80 | 51.82 | 0.18 | 0.19 |
| 03321500 | 9,181 | 112 | 3.03 | −1.50 | −1.63 | −1.52 | −1.56 | 135.52 | 86.97 | 14.81 | 0.12 | 0.09 |

| 03374100 | 11,305 | 123 | 0.52 | −1.54 | −1.51 | −1.20 | −1.13 | 142.22 | 106.97 | 37.06 | 0.09 | 0.08 |
|---|---|---|---|---|---|---|---|---|---|---|---|---|
| 06037500 | 435 | 2,026 | 0.00 | −1.50 | −1.52 | −0.31 | −0.29 | 97.97 | 88.25 | 30.00 | 0.18 | 0.17 |
| 06228000 | 2,309 | 1,504 | 1.01 | −1.64 | −1.28 | −1.13 | −0.76 | 286.39 | 198.74 | 6.05 | 0.11 | 0.13 |
| 06296120 | 42,847 | 712 | 1.19 | −1.42 | −1.35 | −0.55 | −0.49 | 268.60 | 263.99 | 25.19 | 0.17 | 0.19 |
| 06340500 | 2,240 | 530 | 0.75 | −1.29 | −1.31 | −0.91 | −0.89 | 623.07 | 324.24 | 104.73 | 0.07 | 0.06 |
| 06892350 | 59,756 | 242 | 1.47 | −1.84 | −1.99 | −0.92 | −0.94 | 453.80 | 536.11 | 78.94 | 0.18 | 0.25 |
| 07075250 | 48 | 270 | 0.61 | −1.33 | −1.22 | −3.65 | −3.26 | 39.66 | 33.64 | 3.84 | 0.24 | 0.24 |
| 07075270 | 75 | 214 | 10.83 | −1.49 | −1.46 | −5.19 | −5.05 | 27.54 | 25.96 | 1.29 | 0.08 | 0.08 |
| 07079300 | 50 | 3,026 | 5.47 | −1.56 | −1.55 | −1.37 | −1.34 | 106.11 | 96.28 | 27.54 | 0.11 | 0.10 |
| 07081200 | 99 | 2,955 | 0.46 | −1.39 | −1.43 | −1.08 | −1.09 | 41.91 | 45.81 | 6.92 | 0.06 | 0.06 |

i.e. 1, 2 represents yearly dynamic max and yearly dynamic 99th respectively

### 3.3 Data requirements for determining $SC_{BF}$ and $SC_{RO}$

The $SC_{BF}$, $SC_{RO}$ and BFI values tended to stabilize with increasing sampling duration. In general, with a gradual increase in $SC_{BF}$, $SC_{RO}$ showed a decreasing trend, whereas BFI showed fluctuation with no significant upward or downward trend (e.g. stream site 07086000 shown in Fig. 4 and other sites shown in Supplement 1). The P values of BFI as determined by the T-test did not indicate significant changes with sampling duration, which were greater than 0.05 for durations longer than 3 months. The uncertainty of BFI (i.e. $W_{BFI-Genereux}$) similarly showed significant variation of as high as 0.31 at a conductivity sampling duration of 3 months, but stabilized in the range of 0.14 to 0.27 for sampling duration greater than 3 months (Fig. 5). Therefore, it is clear that a BFI obtained from any continuous data with a sampling duration no longer than three months will obviously differ from that obtained from data with a two-year continuous sampling duration. Therefore, at least six months of conductivity records are suggested to obtain reliable estimates of $SC_{BF}$, $SC_{RO}$ and BFI. Stream sites in which the BFI followed a normal distribution (~20 stream sites) were assessed, and it was found that there were 10 sites with minimum sampling duration of 3 months and 6 months, respectively (see Supplement 1 and 2 for details). Therefore, a minimum of 6 months sampling duration is recommended for application of the CMB method to separate the hydrograph for sites in the Mississippi River Basin.

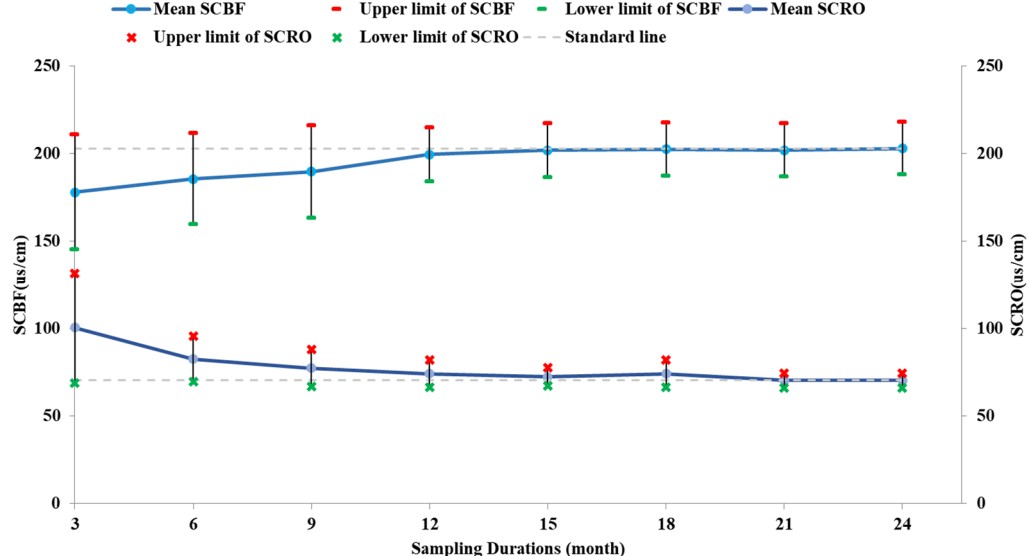

**Figure 4. Values and variations of SC$_{BF}$ and SC$_{RO}$ with different sampling durations (error bars indicate ± one standard deviation for each sampling duration).**

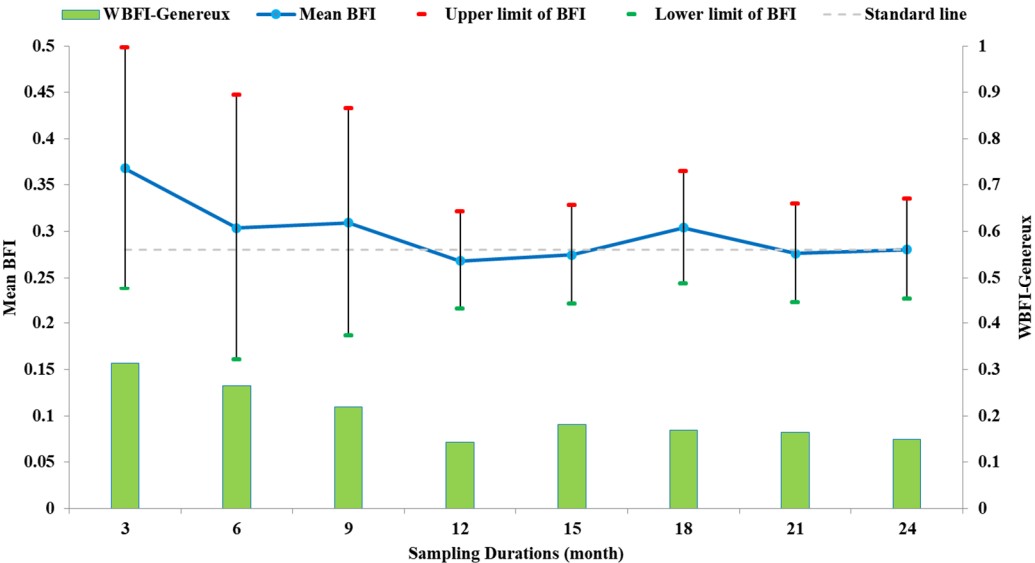

**Figure 5. Values and variations of Mean BFI and W$_{BFI-Genereux}$ with different sampling durations**

## 4. Discussion

### 4.1 Sub-basin characteristics as indicators of the applicability of the CMB method

The results of the present study suggested that the applicability of the CMB method to a particular site can be determined

by the presence of an inverse correlation between streamflow discharge and conductivity within monitoring data. Baseflow separation showed unreasonable results for sites in which there was no significant inverse correlation between stream conductivity and discharge. Taking site 01636315 as an example (Fig, 6), an increase in river flow from 28 August, 2006 to 16 December, 2006 was accompanied by a consistently high level of conductivity over the entire monitoring period. The calculated baseflow for this site using Eq. (1) was too large, with a significantly higher ratio during the flood process which clearly did not conform with the mechanism of the baseflow recharge process. During periods of recession (for example, 23 July, 2007–06 November, 2007, 09 June, 2008–24 August, 2008, 30 June, 2009–21 October, 2009, 23 May, 2010–11 August, 2010), a gradual decrease in discharge was accompanied by a gradual decrease in conductivity, which is an opposite trend in what would be expected, and resulting in the calculated baseflow hydrograph being significantly lower than the runoff hydrograph. During the dry season, the only source of water in the river was baseflow, and therefore the separation results were clearly incorrect. In fact, for sites in which there was no significant inverse correlation between stream discharge and conductivity, they tended to show a positive relationship. Under these conditions, baseflow separation will generate inaccurate baseflow estimates. Therefore, the present study confirmed the value of an inverse correlation between conductivity and discharge as an indicator of the suitability of the CMB method.

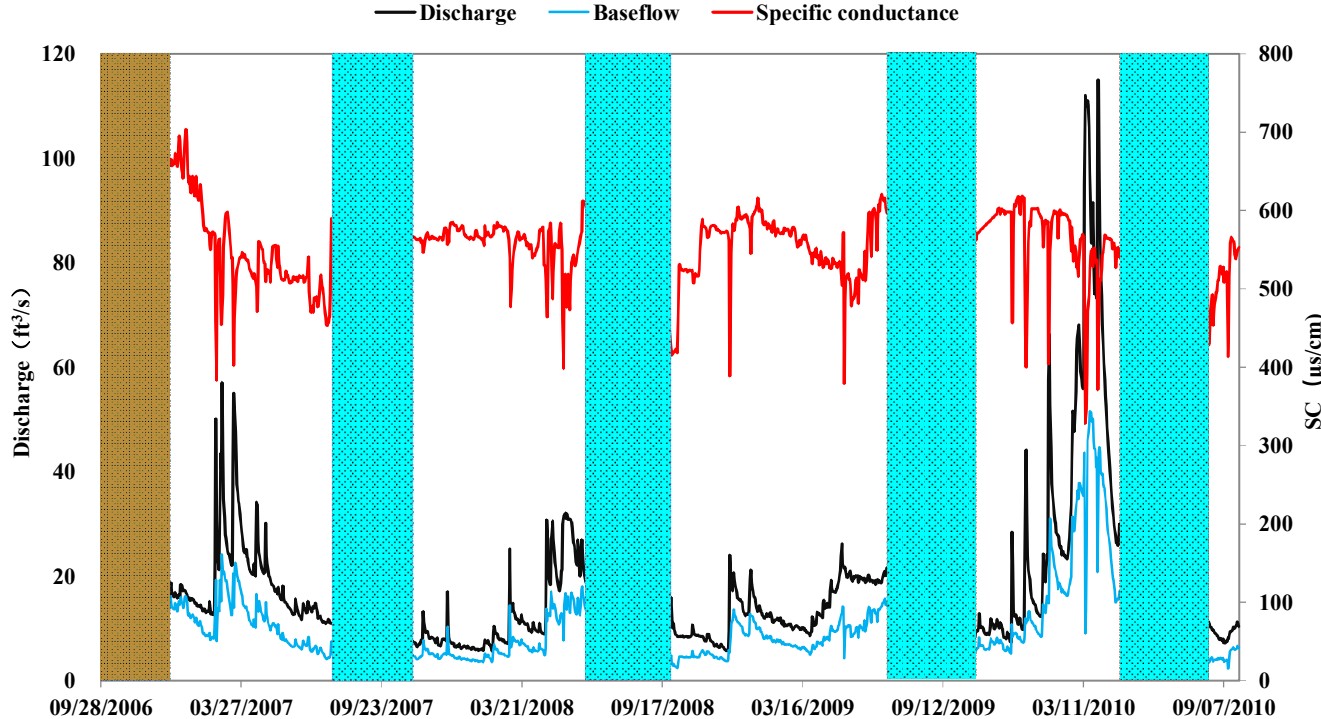

**Figure 6 Temporal variation in discharge, specific conductance and baseflow for a typical site in the Mississippi River Basin**

The presence of an inverse correlation between stream conductivity and discharge is dependent on a strong hydraulic connection between groundwater and surface water in a reach and on the major direction of surface water-groundwater interaction being from groundwater to surface water. The CMB method should not be applied to sites in which there is interference in this relationship through anthropogenic activities and other external factors. In this way, conductivity and streamflow data can accurately reflect the natural spatial and temporal variation in baseflow and in the baseflow index. The present study further analyzed the characteristics of factors influencing the inverse correlation between stream conductivity and discharge, including location, topography, surrounding environmental conditions and anthropogenic interferences. By combining the inverse correlation and baseflow separation results, the present study provides a discussion of the key factors influencing the applicability of the CMB method.

(1) Impacts of topography and altitude

More than 90% (18/19) of the sites located in the upstream area of the basin characterized by a steep terrain and high altitude (particularly those above 1,500 m) showed an inverse correlation (i.e. $r \leq -0.5$) between streamflow conductivity and discharge, thereby indicating the good applicability of the CMB method for these sites (Fig, 7). In these areas, high flow velocity and a significant downcutting effect of the river contribute to V-shaped river valleys. There is a strong hydraulic connection between groundwater and surface water in these cases. The middle and lower river reaches are in contrast characterized by lower flow velocity, a weakened downcutting effect, and as the river water level rises, the river may cross a threshold in which it becomes a source of groundwater recharge. This change in relationship between surface water and groundwater results in a breakdown in the inverse correlation between conductivity and discharge, thereby violating the mechanistic understanding the CMB method is based on. In particular, the lower reaches of the basin downstream of Cairo are characterized by a reduced riverbed gradient, wider river valleys and circuitous river channels in which groundwater is recharged by surface water and the ratio of sites with a medium to high degree of inverse correlation (i.e. $r \leq -0.5$) is reduced to 55% (101/182), suggesting that the applicability of CMB method for these sites is significantly reduced. As shown in Fig. 8, the proportion of sites with a correlation coefficient less than −0.5 increased significantly with increasing site elevation. However, the relationship between the correlation coefficient and site elevation did not strictly satisfy linear inverse correlation, and there are also some sites below 1,500 meters (especially 500 meters) that met the requirements of the correlation coefficient (less than −0.5), these sites were mainly located in the Ohio River Basin, the terrain of the basin is relatively flat and the elevation is low. Since the elevations of many sites located in stream headwater areas were less than 500 m, the impact of site location (such as on a tributary or main stem) may have be more significant than elevation.

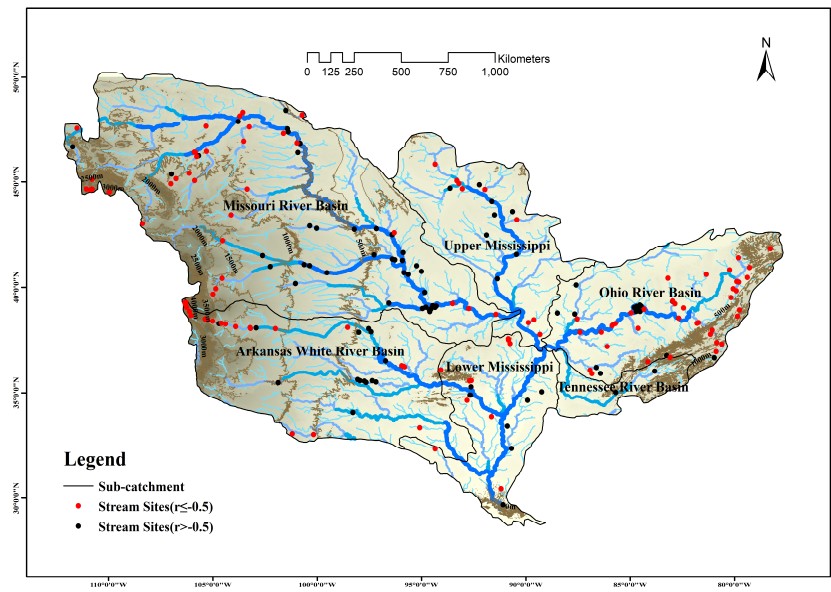

**Figure 7. Ground elevation and spatial distribution of correlation coefficients for the correlation between stream conductivity and discharge in the Mississippi River Basin**

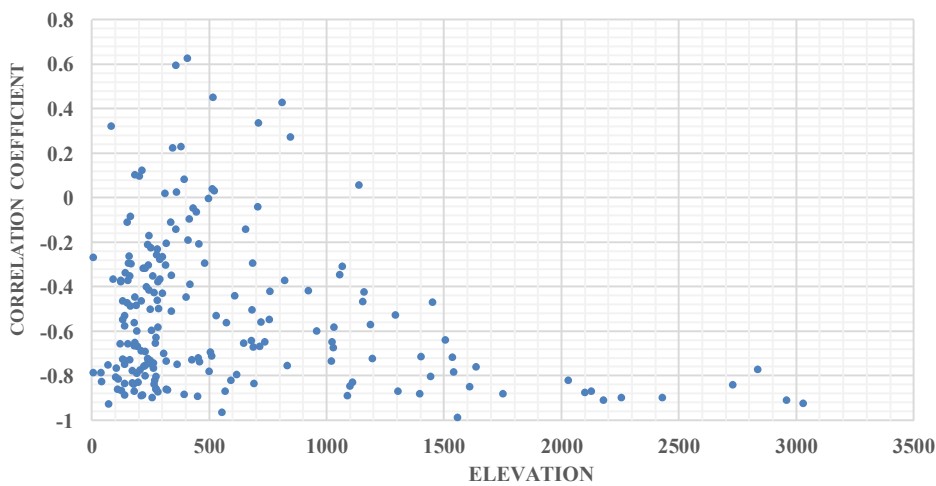

**Figure 8. Scatterplot of the correlation coefficient against the elevation of the Mississippi River Basin monitoring sites**

(2) Impacts of site location and streamflow discharge

The present study analyzed and compared site data for the main stem and tributaries of the Missouri River Basin, Arkansas River Basin, upper Mississippi River Basin and other sub basins. The results showed that a higher proportion of sites in the tributaries met the requirements of the CMB method. For example, the proportions of tributary and main stem sites which met the requirements of the CMB method in the Missouri River, Ohio River and upper Mississippi River were 51.6% and 36.4%,

70.5% and 50%, and 54.5% and 50%, respectively. Tributaries sites were generally characterized by a high-altitude and steep terrain, whereas the mainstem sites fell within plain and low-altitude areas. Therefore, in general, the CMB method is more likely to be applicable to tributary sites.

In theory, streamflow discharge should be a strong determinant of the feasibility of the CMB method. Within a specific watershed, sites with high discharge are mostly located along the mainstems and downstream area, and as discussed above, few are suitable for application of the CMB method. On the other hand, sub-basins with lower flow are likely to be more susceptible to temporal variations in water quantity and the influences of external factors, resulting in distorted results of baseflow separation. However, the results of the present study showed no consistent mathematical relationship between streamflow discharge and correlation coefficient r. Considering the existence of a strong linear relationship between discharge and catchment area for certain sub basins, for example for the Missouri River Basin in which the $R^2$ of the relationship is 0.94, further analysis on the relationship between catchment area and the applicability of the CMB method was justified. c The present study found that the proportion of monitoring sites with a strong inverse correlation coefficient for the stream conductivity-discharge relationship (i.e. $r \leq -0.5$) was relatively low under a very large catchment area. For example, within the Arkansas River Basin, only ~11% of sites with an area > 34,000 km$^2$ showed a strong inverse correlation coefficient (Fig, 9a). In addition, the proportion of monitoring sites with catchment areas < 800 km$^2$ in which there was a strong inverse correlation coefficient (i.e. $r \leq -0.5$) was relatively low, with approximately 20% in the Missouri River Basin (Fig, 9b). However, it is difficult to simultaneously determine the high-flow and low-flow thresholds for applicability of the CMB method within a particular sub basin.

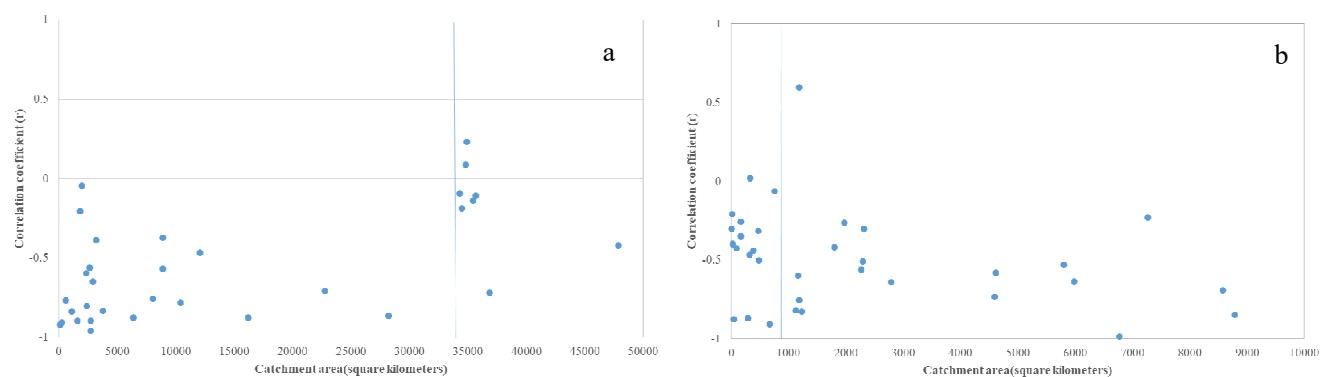

**Figure 9 Catchment area and correlation coefficient of each site in the Mississippi River Basin**

(3) Impacts of anthropogenic factors

Human activities can significantly affect stream discharge and water quality, thereby disrupting their natural relationship and invalidating the application of the CMB method. Human activities can result in dramatic changes to river conductivity, and the major impact processes include agricultural irrigation, mining activity, and the use of salts as road de-icing agents (Kaushal et al., 2005; Crosa et al., 2006; Dikio, 2010; Palmer et al., 2010; Bäthe and Coring, 2011; Miguel et al., 2013). Other anthropogenic factors can also result in artificial variations in conductivity, such as industrial wastewater discharge (Piscart et

al., 2005b; Dikio, 2010), discharge of sewage wastewater (Silva et al., 2000; Williams et al., 2003; Lerotholi et al., 2004) or reduced river discharge due to river impoundment (Mirza, 1998).

Irrigation and the resulting rise in groundwater tables has been reported as one of the main factors leading to significant changes in electrical conductivity of river water, particularly in arid and semi-arid regions in which crop production consumes large quantities of water. Since crops absorb only a fraction of salt introduced through irrigation water, the remaining salt concentrates in the soil, leading to saline soil (Lerotholi et al., 2004). These salts may be leached out through run-off, ultimately ending up in rivers. Therefore, agriculture practices such as fertilizer application can influence the concentrations of conductivity and hence affect the accuracy of the CMB method. In contrast, Li et al. (2018) showed that conductivity of baseflow and surface runoff did not change over time in forest watersheds.

Mining activity is another major source of salts in rivers. Large quantities of potash salts are extracted each year for the manufacture of agricultural fertilizers. During the process of manufacturing of crude salt, which contains not only potash, but also NaCl and other salts, huge amounts of solid residues are stockpiled. The salts are dissolved during precipitation events and may enter surface waters. Mountaintop mining is a mining technique which involves removing 500 or more feet of a mountain to gain access to coal seams, and has been blamed for large-scale stream salinization (Pond et al., 2008). The exposure of coal seams to weathering and percolation during coal mining provides many opportunities for the leaching of sulphate from coal wastes into surface waters (Fritz et al., 2010; Bernhardt and Palmer, 2011).

Significant changes in electrical conductivity in the cold regions has often been often reported to be the result of the use of salts as road de-icing agents (Löfgren, 2001; Ruth, 2003; Williams et al., 2003). The amount of salts used to de-ice roads in North America increased from 909,000 to 1,347,000 tons per winter from 1961 to 1966 (Hanes et al., 1970). During the 1980s, the amount of salts applied to roads increased to 10 million tons per year in the United States alone (Salt Institute, 1992). Around 14 million tons of salt per year is currently applied to roads in North America (Environment Canada, 2001). The majority of salts used on roads are transported to adjacent streams during rainfall events and snow melting periods (Williams et al., 2003). Consequently, concentrations of salts downstream from major roads have been recorded to be up to 31 times higher than comparative upstream concentrations (Demers and Sage, 1990) and some rural streams have registered chloride concentrations exceeding $0.1 \text{ g L}^{-1}$ ($\approx 0.16 \text{ g NaCl g L}^{-1}$), similar to those found in the salt front of the Hudson River estuary (Kaushal et al., 2005).

Typically, a monitoring site is located adjacent to a reservoir or other water conservancy infrastructure, which may contribute to significantly increased evaporation and higher conductivity. On the other hand, the reservoir/dam can also provide substantial sources of water in low flow periods. This may decrease conductivity in streams, thereby undermining the groundwater contribution to streams and leading to an underestimation of baseflow conductivity. In the present study, such affected stream sites included 07130500, 05116000, 06058502, 03400800 and 05370000 located in the upstream part of the Mississippi River Basin, and these sites showed relatively poor inverse correlations between stream conductivity and discharge, with the correlation coefficients of −0.42, −0.29, 0.06, −0.44 and −0.495, respectively.

Since the Mississippi River Basin encompasses almost 2/3 of the entire area of the United States and streamflow occurs

through large areas of plain in the Midwest and densely populated areas in the east, the impacts of anthropogenic factors in these areas are great, resulting in limited applicability of the CMB method.

The present study found that in general, for the entire Mississippi River Basin, the CMB method was more applicable for headwater sites, tributaries and high-altitude regions of > 1,500 m above sea level, with relatively little impacts by anthropogenic factors. In contrast, the application of the CMB method to downstream flat and low-altitude areas or to areas affected by anthropogenic activities should be carefully considered.

A related study in the Upper Colorado River Basin suggest higher elevation watersheds typically have greater baseflow yield (Rumsey et al., 2015), and Dyer (2008) found that high flows in upper streams are mainly stimulated by the snow-melt process. TWhether the impacts of altitude and site location are mainly due to differences in hydrological regimes, i.e., snow-dominated in upper streams and rain-dominated in lower watersheds. From these findings which are based on the major river basins in North America, we still can't establish a relationship between hydrological regimes and the applicability of CMB method. On the other hand, as a large watershed, the Mississippi River basin has sizeable spatial heterogeneity of climate. The role of climate on hydrology, particularly for low flows, is more pronounced in larger watersheds. The influence of hydrological processes on baseflow is complex, particularly when taking climate change into consideration. Therefore, specialized research will be required in the future.

## 4.2 Optimal method to determine $SC_{BF}$ and $SC_{RO}$

The comparison of sensitivity analysis results indicated that the influence of parameter $SC_{RO}$ on the separation results was significantly lower than that of parameter $SC_{BF}$. This result is supported by previous relevant research (Stewart et al., 2007; Zhang et al., 2012; Li et al., 2014; Yang et al., 2019). Moreover, since $SC_{RO}$ represents the minimum conductivity during the wet season whereas $SC_{BF}$ represents the maximum conductivity during the dry season, the $SC_{RO}$ is less likely to be reduced to an unreasonable extremely low value by the effects of natural or anthropogenic activities. The present study conservatively recommends the 99th percentile of conductivity of the entire monitoring period as indicative of the $SC_{RO}$ to avoid extreme values.

Over a long-term monitoring period, river water quality is often influenced by anthropogenic processes such as release of water from upstream reservoirs and sewage discharge, which can result in extremely high conductivity. Under these situations, taking the maximum conductivity as $SC_{BF}$ will result in inaccurate baseflow separation, and the use of the 99th percentile of conductivity can effectively avoid these extreme situations. Considering that the climate, human activities and corresponding hydrological processes occurring in a basin will change greatly over the full extent of a monitoring period, it is recommended that the $SC_{BF}$ be determined dynamically to further improve the accuracy of baseflow separation. From the calculated uncertainty results of each method (Table 1), it can be concluded that the uncertainty associated with the use of the dynamic 99th percentile approach was lower than that of the dynamic maximum conductivity approach. Taking site 07097000 as an example for comparing the four approaches of assigning $SC_{RO}$ and $SC_{BF}$ (Fig. 10), during the recession process, the baseflow calculated by the recommended approach appeared rational, whereas the other three approaches generated relatively low

baseflow. Therefore, it is suggested that the 99th percentile of conductivity of the entire monitoring period and yearly dynamic 99th percentile approach should be used to determine $SC_{RO}$ and $SC_{BF}$, respectively.

However, it must be stressed that although the applicability of the CMB method has been verified for a site before determining parameters, it cannot be guaranteed that there will be no anthropogenic disturbance to parameters of a site in which the CMB method has been found to be applicable, and that the parameters correspond to the lowest flows very well. For example, leakage of an underground storage tank may last for a long time, which may result in many observations of extremely high conductivities that cannot be avoided by the 99th percentile method. So there is a possibility that the 99th percentile conductivity does not correspond to lowest flows. Therefore, parameters should assessed after calculation by the 99th percentile method to further avoid abnormal phenomena and errors within separation results.

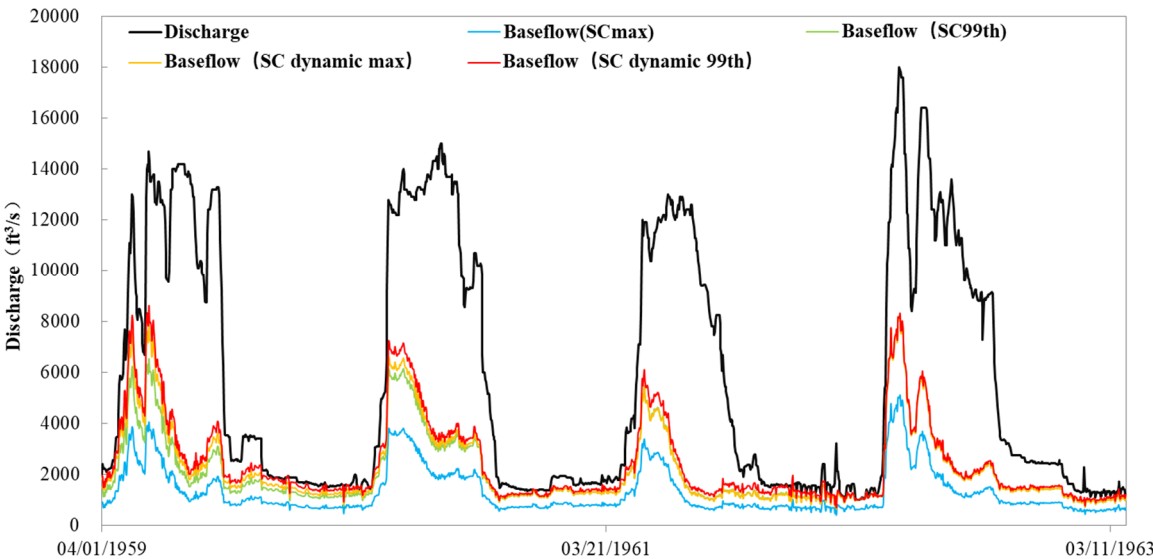

**Figure 10 Comparison of baseflow calculation results of main parameter determination methods for a site (07097000) in the Mississippi River Basin**

**4.3 Data requirements for $SC_{BF}$ and $SC_{RO}$**

Determining the shortest monitoring periods appropriate for calculating $SC_{RO}$ and $SC_{BF}$ requires determining the monitoring period required to obtain the reference standard of separation results. Generally, the length of the monitoring period is positively related to the accuracy of the hydrological characteristics of the station reflected by the monitoring data, and the BFI result obtained from a longer monitoring record will be more reasonable compared to that obtained from a relatively shorter record. As an example in the present study and using the BFI calculated by 24 months of data as a standard, the random selection of 20 segments in which no more than half of the data were reused will require monitoring periods of greater than 21 years. For this reason, only 26 of 201 sites were selected for analysis in the present study, from which 5 sites allowed the

standard BFI calculation from 24 months of data whereas the remaining 21 sites allowed the BFI to be calculated from 12 months of data. Therefore, there needs to be further comparison and discussion of the data requirements of utilizing different standard sampling durations. The BFI calculated from 24-month data and yearly data were viewed as standard for the four stream sites in which the standard sampling durations were 24-months and in which the monitored data followed a normal distribution, respectively. The student's T-test was used to compare differences in BFI obtained from 3, 6 or 9 months of data and the BFI obtained from standard sampling durations (Table 2). The results showed that minimum sampling duration were all less than or equal to 6 months, which indicated that the results obtained by 12 months sampling duration as a standard were also reasonable. Li et al. (2014) similarly questioned their assumption of requiring a dataset of 12 months duration to provide the best representativeness for a watershed and stressed that the uncertainties associated with variations in $SC_{RO}$ and $SC_{BF}$ over years require further study. The results of the present study support their hypothesis that variations in $SC_{RO}$ and $SC_{BF}$ over years will not have a substantial impact on the determination of standard sampling duration.

**Table 2 Differences between the baseflow index (BFI) obtained from 3, 6, or 9 months data and the BFI obtained from standard sampling durations**

| Site number | Standard sampling duration | Sampling duration | | |
|---|---|---|---|---|
| | | 9-month | 6-month | 3-month |
| 06711565 | 24-month | 0.860 | 0.092 | 0.000 |
| | 12-month | 0.734 | 0.326 | 0.003 |
| 07086000 | 24-month | 0.447 | 0.591 | 0.040 |
| | 12-month | 0.279 | 0.414 | 0.021 |
| 06089000 | 24-month | 0.930 | 0.939 | 0.024 |
| | 12-month | 0.507 | 0.440 | 0.123 |
| 07097000 | 24-month | 0.313 | 0.189 | 0.752 |
| | 12-month | 0.642 | 0.419 | 0.980 |

## 5. Conclusions

Through comprehensive qualitative and quantitative analysis of stream discharge and conductivity data for more than 200 hydrological stations in the Mississippi River basin, the present study systematically addressed key questions related to the application of the CMB method to particular sites for baseflow separation. In general, the CMB method was found to be more applicable to tributaries, headwater sites, sites at high altitude and sites with little influence from anthropogenic activities. The applicability of the CMB method can be determined by analyzing the inverse correlation between stream discharge and conductivity. Continuous monitoring of flow and conductivity of longer than 6 months duration is required to ensure the reliability of baseflow separation results within the CMB method. Within a long series of monitoring data, the 99th percentile method and dynamic 99th percentile method are recommended to determine the parameters of $SC_{BF}$ and $SC_{RO}$, respectively.

Further study is required to determine which 6 months should be selected for continuous monitoring after the shortest sampling period is determined, as this could be closely related to the geographical location and meteorological conditions of each station. In addition, future research should address whether monitoring should occur during the wet season, dry season or both. Future research should also consider large watersheds in other latitudes and climates so as to compare and verify the conclusions of the present study and to establish more generalized methods. The present study can act as a reference for the

identification of parameters of baseflow separation methods so as to improve the accuracy of these methods.

**Data availability**

All streamflow and conductivity data can be retrieved from the US Geological Survey's (USGS) National Water Information System (NWIS) website using the special site number: http://waterdata.usgs.gov/nwis (NWIS, 2018).

**Author contributions**

HL developed the research train of thought. CX completed the data requirement analysis. JZ carried out the CMB method suitability assessment. BL compared different parameter determination methods. HL prepared the manuscript with contributions from all coauthors.

**Competing interests**

The authors declare that they have no conflict of interest.

**Acknowledgements**

This work is supported by the project funded by National Key R&D Program of China (2018YFC0406503) and the National Natural Science Foundation of China (U19A20107, 41702252), China Postdoctoral Science Foundation (149194). We would like to express our sincere thanks to the editor and the anonymous reviewers for the constructive and positive advice and comments which helped improve the manuscript.

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
