# Peer review of "Key challenges facing the application of the conductivity mass balance method: a case study of the Mississippi River Basin"

_Hydrology and Earth System Sciences, 2020_

## Referee Comment (RC1) · Anonymous Referee #1 · 12 Aug 2020

The conductivity mass-balance (CMB) method has been widely applied to baseflow separation studies for years. But there are some issues have not yet been standardized. This manuscript presents an detail study on the issues which hindering the application of the generally accepted conductivity mass-balance baseflow separation method. I think the results may have a substantial contribution on the standardized treatment of key problems in the application of the CMB and the paper can be accepted by minor revision. A few comments and suggestions are listed below.

1. In Line133, page 5, it has mentioned that "assigning the 99th percentile (ordered

[Figure]

by increasing conductivity) of the stream conductivity monitoring record to avoid the impacts of extremely high SCBF estimates on the separation results", please indicate which conditions can cause extremely parameter values?

2. The study has applied both the uncertainty estimation methods of BFI proposed by Yang et al. (2019) and Genereux and Hooper (1998) to determine the parameters and the shortest time series in the present study. Why do we use both methods at the same time and what are the differences between themïïj§

3. In table1, why not compare the uncertainty results of the various WSCRO determination methods?

4. 2. Fig. 1,3,6,7 should be replaced by more clearer pictures.

5. In the conclusion part, it is suggested that large watersheds in other latitudes and climates should be considered in the future research, so as to compare and verify the conclusions of this study, and to obtain more general guiding methods.

6. In the future research, it is suggested that the results of this method can be used to identify the parameters of other methods to improve the accuracy of separation results of other methods

7. Reference format is not consistent. It should follow the guidelines of the Journal.

---

## Referee Comment (RC2) · Anonymous Referee #2 · 22 Aug 2020

This manuscript discussed the key challenges for applying the conductivity mass balance (CMB) method for baseflow separation and recommend guidelines for the method, which significantly augment the user confidence in applying the CMB for baseflow separation. This work is timely, given that the preliminary literature is lacking the sufficient knowledge in this research theme. Authors adopted large dataset and tests in the Mississippi River Basin to conclude. However, some uncertainties need to be addressed before publication. I, therefore, recommend moderate revision for the current version of the manuscript.

1. In this study, more than 200 sites were included in the data analysis. However, some conclusions were drawn from the simple examination, which lacks the robust evidence of whether these conclusions will hold. For instance, A) the impacts of topography and altitude (Line 276) is concluded by a simple spatial plot (figure 7). In my view, such suggestion is acceptable, but not robust. I suggest the authors can make a scatter plot the correlation against the median elevation of sub-watersheds or other indices that can represent the watershed topography. B) Impact of anthropogenic factors. In this section, the authors only discussed the reservoir as an indicator of human interruption. Disapprovingly, authors only mention the evaporation. The reservoir/dam can provide substantial sources of water in the low flow periods. This may decrease the conductivity in streams and hence undermines the groundwater contribution to streams and leads to an underestimate of baseflow conductivity. Besides, there are other anthropogenic factors such as groundwater pumping and agriculture activities that affect the conductivity in streams and should be discussed in the manuscript.

2. The authors stressed that there is a large amount of watershed where CMB can not be applied. The following question is why this happens in this watershed? I assume that further tests are needed to answer this question. Based on my experience, I suggest the authors can test, but not limited to, the following variables: A) watersheds area, B) Watershed locations, C) snow- and rain-dominated hydrological regimes, D) Land cover and land use. E) Climate regions.

2.1 In Figure 8, authors only examined the relationship between correlation and watershed area in two sub-watersheds. Why don't you examine such relationship for all study watersheds? In smaller watersheds, low flows are mainly fed by groundwater. In contrast, there is always a large amount of surface runoff in the low flows period due to the spatial heterogeneity of climate. In my opinion, you could test all sub-watersheds as well as the entire Mississippi River Basin, and it can drive a threshold of watershed area, above which the CMB methods cannot be applied.

2.2 In this study, the authors concluded that headwater watersheds have a better relationship between discharge and conductivity. I assume this is likely due to differences in hydrological regimes, i.e., snow-dominated and rain-dominated. In upper streams, high flows are mainly stimulated by the snow-melt process (e.g., Dyer, 2008). They can be classified as snow-dominated watersheds, while lower watersheds are more likely to be rain-dominated systems. Two systems have a distinct hydrological process, and there is potential uncertainty whether there is a significant difference between the two systems.

Dyer, J., 2008. Snow depth and streamflow relationships in large North American watersheds. Journal of Geophysical Research: Atmospheres, 113(D18).

2.3 Land cover and land use can be a factor. Forest cover and agriculture land use can have different conductivity concentrations. In the forest watersheds, Li et al. (2018) (in supplementary) showed that conductivity of baseflow and surface runoff did not change over time. In contrast, agriculture practices such as fertilizer application can influence the concentrations of conductivity and hence affect the CMB method accuracy.

2.4 Mississippi River basin is the large watershed. The basin has sizeable spatial heterogeneity of climate. The role of climate on hydrology, particularly for low flows are more pronounced in the larger watersheds. It is worth conducting an analysis of this topic. For simplicity, Climate North America (http://climatena.ca/) can provide climate data for the basin.

In sum, further analysis is, for sure, needed to address the knowledge gaps as mentioned above.

Li, Q., Wei, X., Zhang, M., Liu, W., Giles-Hansen, K. and Wang, Y., 2018. The cumulative effects of forest disturbance and climate variability on streamflow components in a large forest-dominated watershed. Journal of Hydrology, 557, pp.448-459.

3. One recommendation of this manuscript is that the parameters SCro and SCbf can be determined by the 99th percentile and dynamic 99th percentile methods. I

agree with the authors to select the 99th percentile of conductivity. However, there is also a concern related to this recommendation. For the CMB method, the SCbf is often corresponding to the lowest flows with potential time lags (Li et al. 2014; in your manuscript). With the recommendation of using the 99th percentile, it might be a chance that the 99th percentile does not correspond to the lowest flows. Therefore, this should be mentioned in the discussion.

4. The title can be rephrased as "Key challenges facing the application of the conductivity mass balance method: a case study of the Mississippi River Basin"

5. Table 1 should be reorganized. It is meaningless to use the site number. I suggest the site characteristic such as watershed area, relief, slope, and climate, can also be listed in table 1. As such, the sensitivity can be compared with watershed characteristics.

6. The objective should be concise. In Line 85, "to resolve some of the questions". Please be more specific, which questions you are going to resolve in this manuscript.

7. Section 2.5, the meaning of the sensitivity and uncertainty should be elaborated more. For instance, larger values of sensitivity indicate higher sensitivity. A similar explanation is needed for uncertainty.

8. The language should be polished by the professionals before publication. Here I list some of the suggestions while I read the manuscript.

i. First two sentences in the abstract. Suggestion: The conductivity mass balance (CMB) method has a long history of application to baseflow separation studies, which uses site-specific and widely available discharge and specific conductance data.

ii. Line 17, insert "in" ; the parameter in the method

iii. Lines 45-47, rewrite

iv. Line 125, the key parameters need to be calculated

v. Line 140, for at least 5 years

vi. Line 147, delete unbroken

vii. Figure 8 is not clear. Please redraw.

All in all, the above mentioned are the suggestions for this manuscript. I am looking forward to your revision.

---

## Author Comment (AC2) · 4 Sep 2020

We highly appreciate Anonymous Referee 2 for extensive and generous comments on the manuscript and his/her generally positive impression of our work. Here we briefly respond to the points raised by his remarks.

[Figure]

1.In this study, more than 200 sites were included in the data analysis. However, some conclusions were drawn from the simple examination, which lacks the robust evidence of whether these conclusions will hold. For instance, A) the impacts of topography and altitude (Line 276) is concluded by a simple spatial plot (figure 7). In my view, such suggestion is acceptable, but not robust. I suggest the authors can make a scatter plot the correlation against the median elevation of sub-watersheds or other indices that can represent the watershed topography. B) Impact of anthropogenic factors. In this section, the authors only discussed the reservoir as an indicator of human interruption. Disapprovingly, authors only mention the evaporation. The reservoir/dam can provide substantial sources of water in the low flow periods. This may decrease the conductivity in streams and hence undermines the groundwater contribution to streams and leads to an underestimate of baseflow conductivity. Besides, there are other anthropogenic factors such as groundwater pumping and agriculture activities that affect the conductivity in streams and should be discussed in the manuscript.

Author's response: Thank you for your advice. We have made a scatter plot of the correlation against the elevation of sites (Fig.1), it can be found that with the increase of site elevation, the proportion of sites with a correlation coefficient less than -0.5 increased significantly, and it's consistent with our previous conclusions (most stations located in stream headwater areas, with an elevation above 1,500 m, a steep terrain and high elevation showed inverse correlations between flow and conductivity). However, the relationship between them does not strictly satisfy the inverse correlation, and we also found that there are also many sites below 1500 meters (especially 500 meters) that meet the requirements of the correlation coefficient (less than -0.5), these sites mainly located in the Ohio River Basin, the terrain of the basin is relatively flat and the altitude is low, the elevation of many sites located in stream headwater areas are still less than 500 meters, so the impact of elevation in this sub-watershed is not significant. More robust evidence will be added in the revised manuscript.

Impact of anthropogenic factors. We researched the relevant literature again. As the

Referee mentioned, "there are other anthropogenic factors that affect the conductivity in streams and should be discussed in the manuscript." We found that changes in river conductivity can have many different causes, and the major impact processes include agriculture practices (such as fertilizer application), mining activity, the use of salts as deicing agents for roads (Miguel et al., 2013; Crosa et al., 2006; Palmer et al., 2010; Bäthe and Coring, 2011; Dikio, 2010; Kaushal et al., 2005). Besides, other anthropogenic factors such as discharge from industrial activities (Piscart et al., 2005b; Dikio, 2010), sewage treatment plant effluents (Silva et al., 2000; Williams et al., 2003; Lerotholi et al., 2004) or reduced river discharge due to damming (Mirza, 1998) can also cause the variation of conductivity. Admittedly, previous understanding of the impact of anthropogenic factors is one-sided. The influence of reservoir/dam will be revised and other factors will be discussed in the manuscript.

Miguel Cañedo-Argüelles, Ben J. Kefford, Christophe Piscart, Narcís Prat, Ralf B. Schäfer, Claus-Jürgen Schulz. Salinisation of rivers: An urgent ecological issue, Environmental Pollution 173 (2013) 157-167.

Crosa, G., Froebrich, J., Nikolayenko, V., Stefani, F., Galli, P., Calamari, D., 2006. Spatial and seasonal variations in the water quality of the Amu Darya River (Central Asia). Water Research 40 (11), 2237-2245.

Palmer, M.A., Bernhardt, E.S., Schlesinger, W.H., Eshleman, K.N., Foufoula- Georgiou, E., Hendryx, M.S., Lemly, A.D., Likens, G.E., Loucks, O.L., Power, M.E., White, P.S., Wilcock, P.R., 2010. Mountaintop mining consequences. Science 327 (5962), 148-149.

Bäthe, J., Coring, E., 2011. Biological effects of anthropogenic salt-load on the aquatic fauna: a synthesis of 17 years of biological survey on the rivers Werra and Weser. Limnologica 41(2), 125-133.

Dikio, E.D., 2010. Water quality evaluation of Vaal river, Sharpeville and Bedworth lakes in the Vaal region of south Africa. Research Journal of Applied Sciences, Engineering and Technology 2 (6), 574-579.

Kaushal, S.S., Groffman, P.M., Likens, G.E., Belt, K.T., Stack,W.P., Kelly, V.R., Band, L.E., Fisher, G.T., 2005. Increased salinization of fresh water in the northeastern United States. Proceedings of the National Academy of Sciences of the United States of America 102 (38), 13517-13520.

Piscart, C., Moreteau, J.-C., Beisel, J.-N., 2005. Biodiversity and structure of macroinvertebrate communities along a small permanent salinity gradient (Meurthe river, France). Hydrobiologia 551 (1), 227-236.

Silva, E.I.L., Shimizu, A., Matsunami, H., 2000. Salt pollution in a Japanese stream and its effects on water chemistry and epilithic algal chlorophyll-a. Hydrobiologia 437 (1), 139-148.

Williams, M.L., Palmer, C.G., Gordon, A.K., 2003. Riverine macroinvertebrate responses to chlorine and chlorinated sewage effluents e acute chlorine tolerances of Baetis harrisoni (Ephemeroptera) from two rivers in KwaZulu-Natal, South Africa. Water SA 29 (4), 483-487.

Lerotholi, S., Palmer, C.G., Rowntree, K., 2004. Bioassessment of a River in a Semi-arid, Agricultural Catchment, Eastern Cape. In: Proceedings of the 2004 Water Institute of Southern Africa (WISA) Biennial Conference, Cape Town, South Africa, pp. 338-344.

Mirza, M.M.Q., 1998. Diversion of the Ganges water at Farakka and its effects on salinity in Bangladesh. Environmental Management 22 (5), 711-722.

2.The authors stressed that there is a large amount of watershed where CMB can not be applied. The following question is why this happens in this watershed? I assume that further tests are needed to answer this question. Based on my experience, I suggest the authors can test, but not limited to, the following variables: A) watersheds area, B) Watershed locations, C) snow and rain dominated hydrological regimes, D) Land cover and land use. E) Climate regions.

2.1 In Figure 8, authors only examined the relationship between correlation and watershed area in two sub-watersheds. Why don't you examine such relationship for all study watersheds? In smaller watersheds, low flows are mainly fed by groundwater. In contrast, there is always a large amount of surface runoff in the low flows period due to the spatial heterogeneity of climate. In my opinion, you could test all sub-watersheds as well as the entire Mississippi River Basin, and it can drive a threshold of watershed area, above which the CMB methods cannot be applied.

2.2 In this study, the authors concluded that headwater watersheds have a better relationship between discharge and conductivity. I assume this is likely due to differences in hydrological regimes, i.e., snow-dominated and rain-dominated. In upper streams, high flows are mainly stimulated by the snow-melt process (e.g., Dyer, 2008). They can be classified as snow-dominated watersheds, while lower watersheds are more likely to be rain-dominated systems. Two systems have a distinct hydrological process, and there is potential uncertainty whether there is a significant difference between the two systems. Dyer, J., 2008. Snow depth and streamflow relationships in large North American watersheds. Journal of Geophysical Research: Atmospheres, 113(D18).

2.3 Land cover and land use can be a factor. Forest cover and agriculture land use can have different conductivity concentrations. In the forest watersheds, Li et al. (2018) (in supplementary) showed that conductivity of baseflow and surface runoff did not change over time. In contrast, agriculture practices such as fertilizer application can influence the concentrations of conductivity and hence affect the CMB method accuracy.

2.4 Mississippi River basin is the large watershed. The basin has sizeable spatial heterogeneity of climate. The role of climate on hydrology, particularly for low flows are more pronounced in the larger watersheds. It is worth conducting an analysis of this topic. For simplicity, Climate North America (http://climatena.ca/) can provide climate data for the basin. In sum, further analysis is, for sure, needed to address the knowledge gaps as men- tioned above. Li, Q., Wei, X., Zhang, M., Liu, W., Giles-Hansen, K. and Wang, Y., 2018. The cumulative effects of forest disturbance and

climate variability on streamflow components in a large forest-dominated watershed. Journal of Hydrology, 557, pp.448-459.

Author's response: Thanks for your advice and experience. The suggestions and questions you have proposed are very good research topics, which are worthy of our special research from the aspects you mentioned in the future study.

2.1 Watershed area. We did try to obtain a threshold of watershed area aiming at all sub-watersheds, but the results are not ideal. We found that the watershed area in different sub-watershed is very different, the Missouri River basin, for example, has catchments below 9,000 km2 at all stations, while in the Arkansas River basin, there are many catchments greater than 30,000 km2. Therefore, only two sub-watersheds are listed and discussed in the manuscript. And to avoid confusion, other sub-watersheds will be discussed in the revised manuscript.

2.2 Hydrological or hydrogeological regimes? Or both of them? According to the distribution of correlation coefficient, we have made the conclusion that headwater watersheds have a better relationship between discharge and conductivity. As you mentioned, "this is likely due to differences in hydrological regimes, i.e., snow-dominated and rain-dominated". To solve this problem, we have further consulted relevant literature. A similar study in Upper Colorado River Basin suggest that there is typically greater baseflow yield in higher elevation watersheds (Rumsey et al., 2015), and Dyer(2008) found that in upper streams, high flows are mainly stimulated by the snowmelt process. But from these findings which are based on the major river basins in North America we still can't establish a relationship between hydrological regimes and the applicability of CMB method (quantitatively expressed by correlation coefficient between discharge and conductivity in our study). We still have the opinion that hydrogeological conditions are more dominant, there is a strong hydraulic connection between groundwater and surface water due to the erosion in upper streams under natural condition, and that the major direction of surface water-groundwater interaction is from groundwater to surface water. In this way, conductivity and streamflow data can accu-
rately reflect the natural spatial and temporal variation. The differences in hydrological regimes, i.e., snow-dominated and rain-dominated will be discussed in our future study.

Christine A. Rumsey, Matthew P. Miller, David D. Susong, Fred D. Tillman, David W. Anning., Regional scale estimates of baseflow and factors influencing baseflow in the Upper Colorado River Basin. Journal of Hydrology: Regional Studies, 4 (2015) 91–107. Dyer, J., 2008. Snow depth and streamflow relationships in large North American watersheds. Journal of Geophysical Research: Atmospheres, 113(D18).

2.3 Land cover and land use factor. Thank you for your advice, and we are going to attribute the impact of different land cover and land use to anthropogenic factors, we think that CMB method maintains relatively high accuracy naturally, where less impact of anthropogenic activities happen there, such as forest cover land. In contrast, CMB method is relatively poorly applied to agriculture land with more human intervention to the hydrological process. This will be discussed in the revised manuscript.

2.4 The role of climate. Thank you for your advice, as you mentioned, the role of climate on baseflow are pronounced in the larger watersheds, which is a topic worthy of conducting a special research. We produced a superimposed map of climate zoning and the applicability of CMB method (Fig. 2). However, from this figure, it is difficult to summarize the influence rule of climate type on the applicability of CMB method. According to the website provided by the reviewer, we can further obtain meteorological data, and detailed analyze the influence of meteorological factors on the base flow in future studies. In sum, in order to confirm the reasons that some watershed where CMB can't be applied depends on an overall study in the future, thanks again to the anonymous reviewers for their suggestions.

3. One recommendation of this manuscript is that the parameters SCro and SCbf can be determined by the 99th percentile and dynamic 99th percentile methods. I agree with the authors to select the 99th percentile of conductivity. However, there is also a concern related to this recommendation. For the CMB method, the SCbf

is often corresponding to the lowest flows with potential time lags (Li et al. 2014; in your manuscript). With the recommendation of using the 99th percentile, it might be a chance that the 99th percentile does not correspond to the lowest flows. Therefore, this should be mentioned in the discussion.

Author's response: Thank you for asking the question that makes our research more rigorous. We have checked the parameters determined by dynamic 99th percentile method and didn't found the condition that the 99th percentile doesn't correspond to the lowest flows. But we have to admit that although the applicability of CMB method has been verified for a site before determining parameters, we still can't guarantee that a site where CMB method is applicable possesses parameters with no anthropogenic disturbance and corresponds to the lowest flows well. For example, the leakage of underground storage tank may last for a long time which will result in many extremely high conductivities that can't be avoided by 99th percentile method. Based on the above analysis, we will suggest in our revised manuscript that parameters should be checked after calculating by 99th percentile method to further avoid abnormal phenomena.

4. The title can be rephrased as "Key challenges facing the application of the conductivity mass balance method: a case study of the Mississippi River Basin". Author's response: Thank you for your advice and it will be rephrased in our revised manuscript.

5. Table 1 should be reorganized. It is meaningless to use the site number. I suggest the site characteristic such as watershed area, relief, slope, and climate, can also be listed in table 1. As such, the sensitivity can be compared with watershed characteristics. Author's response: Thank you for your advice and the Table will be reorganized in our revised manuscript.

6. The objective should be concise. In Line 85, "to resolve some of the questions". Please be more specific, which questions you are going to resolve in this manuscript. Author's response: The sentence will be modified to be more concise in the revised manuscript.

[Figure]

7. Section 2.5, the meaning of the sensitivity and uncertainty should be elaborated more. For instance, larger values of sensitivity indicate higher sensitivity. A similar explanation is needed for uncertainty. Author's response: Thank you for your advice and the relative explanation will be added in our revised manuscript.

8. The language should be polished by the professionals before publication. Here I list some of the suggestions while I read the manuscript. First two sentences in the abstract. Suggestion: The conductivity mass balance (CMB) method has a long history of application to baseflow separation studies, which uses site-specific and widely available discharge and specific conductance data. Line 17, insert "in" ; the parameter in the method Lines 45-47, rewrite Line 125, the key parameters need to be calculated Line 140, for at least 5 years Line 147, delete unbroken Figure 8 is not clear. Please redraw. Author's response:

Thank you for your correction, all of them will be revised in the manuscript, and the language will be polished by the professionals before publication.
* * *
[Figure]

**Fig. 1.** Figure 1 Scatter plot of the correlation coefficient against the elevation of Mississippi River Basin monitoring sites

**Legend**

— Sub-catchment

• Stream Sites(r>-0.5)

• Stream Sites(r≤-0.5)

Plateau climate

Semi-arid climate

Humid continental climate(cool)

Humid continental climate(warm)

Humid subtropical climate

**Fig. 2.** Figure 2 Climate type and spatial distribution of correlation coefficients for the correlation between stream conductivity and discharge in the Mississippi River Basin

---

## Author Response (AR1)

**Response to the reviews on "Discussion on key challenges facing the application of the conductivity mass-balance (CMB) method: a case study of the Mississippi River Basin"**

**Editor Decision:**

Reconsider after major revisions (further review by editor and referees) (28 Sep 2020) by Stacey Archfield.

**Comments to the Author:**

The manuscript has received two reviews. While the CRB method is a widely applied method, both reviewers agree that the manuscript is an important advance in formalizing the application of the CRB method and addresses important outstanding questions about its applicability.

I find that, in general, the authors proposed revisions address the reviewer comments. I am, therefore, recommending that the authors now implement their proposed revisions. I would also add there are several author comments that need a more complete response and associated changes in the revised paper. This applies to RC1, comments 1-3 and RC2, comments 2.2 and 2.4, where additional text should be added to the manuscript to address these comments. Even if additional analysis is not conducted, they are important points and should be at least noted. Consider using the author responses as a basis for the additional text in the revision.

The revision will likely be sent for review before a final decision is made. I look forward to the revised paper and thank you for this submission to HESS.

**Author's Response to the Editor:**

This file was structured in the follow sequence: (1) comments from referees, (2) author's response to the referees and corresponding changes in the manuscript, (3) Marked-up manuscript. Regarding the changes, a marked-up manuscript version (track

changes in Word, named "HESS 2020.10.09") has been converted into *.pdf and combined with this response has been provided.

**1. Comments from referees**

**Anonymous Referee #1**

The conductivity mass-balance (CMB) method has been widely applied to baseflow separation studies for years. But there are some issues have not yet been standardized. This manuscript presents an detail study on the issues which hindering the application of the generally accepted conductivity mass-balance baseflow separation method. I think the results may have a substantial contribution on the standardized treatment of key problems in the application of the CMB and the paper can be accepted by minor revision. A few comments and suggestions are listed below.

(1) In Line133, page 5, it has mentioned that "assigning the 99th percentile (ordered by increasing conductivity) of the stream conductivity monitoring record to avoid the impacts of extremely high SCBF estimates on the separation results", please indicate which conditions can cause extremely parameter values?

(2) The study has applied both the uncertainty estimation methods of BFI proposed by Yang et al. (2019) and Genereux and Hooper (1998) to determine the parameters and the shortest time series in the present study. Why do we use both methods at the same time and what are the differences between them?

(3) In table1, why not compare the uncertainty results of the various $WSC_{RO}$ determination methods?

(4) Fig. 1,3,6,7 should be replaced by more clearer pictures.

(5) In the conclusion part, it is suggested that large watersheds in other latitudes and climates should be considered in the future research, so as to compare and verify the conclusions of this study, and to obtain more general guiding methods.

(6) In the future research, it is suggested that the results of this method can be used to identify the parameters of other methods to improve the accuracy of separation results of other methods

(7) Reference format is not consistent. It should follow the guidelines of the Journal.

**Anonymous Referee #2**

This manuscript discussed the key challenges for applying the conductivity mass balance (CMB) method for baseflow separation and recommend guidelines for the method, which significantly augment the user confidence in applying the CMB for baseflow separation. This work is timely, given that the preliminary literature is lacking the sufficient knowledge in this research theme. Authors adopted large dataset and tests in the Mississippi River Basin to conclude. However, some uncertainties need to be addressed before publication. I, therefore, recommend moderate revision for the current version of the manuscript.

(1) In this study, more than 200 sites were included in the data analysis. However, some conclusions were drawn from the simple examination, which lacks the robust evidence of whether these conclusions will hold. For instance, A) the impacts of topography and altitude (Line 276) is concluded by a simple spatial plot (figure 7). In my view, such suggestion is acceptable, but not robust. I suggest the authors can make a scatter plot the correlation against the median elevation of sub-watersheds or other indices that can represent the watershed topography. B) Impact of anthropogenic factors. In this section, the authors only discussed the reservoir as an indicator of human interruption. Disapprovingly, authors only mention the evaporation. The reservoir/dam can provide substantial sources of water in the low flow periods. This may decrease the conductivity in streams and hence undermines the groundwater contribution to streams and leads to an underestimate of baseflow conductivity. Besides, there are other anthropogenic factors such as groundwater pumping and agriculture activities that affect the conductivity in streams and should be discussed in the manuscript.

(2) The authors stressed that there is a large amount of watershed where CMB can not be applied. The following question is why this happens in this watershed? I assume that further tests are needed to answer this question. Based on my experience, I suggest the authors can test, but not limited to, the following variables: A) watersheds area, B) Watershed locations, C) snow and rain dominated hydrological regimes, D) Land cover and land use. E) Climate regions.

① In Figure 8, authors only examined the relationship between correlation and watershed area in two sub-watersheds. Why don't you examine such relationship for all study watersheds? In smaller watersheds, low flows are mainly fed by groundwater. In contrast, there is always a large amount of surface runoff in the low flows period due to the spatial heterogeneity of climate. In my opinion, you could test all sub-watersheds as well as the entire Mississippi River Basin, and it can drive a threshold of watershed area, above which the CMB methods cannot be applied.

② In this study, the authors concluded that headwater watersheds have a better rela-tionship between discharge and conductivity. I assume this is likely due to differences in hydrological regimes, i.e., snow-dominated and rain-dominated. In upper streams, high flows are mainly stimulated by the snow-melt process (e.g., Dyer, 2008). They can be classified as snow-dominated watersheds, while lower watersheds are more likely to be rain-dominated systems. Two systems have a distinct hydrological process, and there is potential uncertainty whether there is a significant difference between the two systems.

Dyer, J., 2008. Snow depth and streamflow relationships in large North American watersheds. Journal of Geophysical Research: Atmospheres, 113(D18).

③ Land cover and land use can be a factor. Forest cover and agriculture land use can have different conductivity concentrations. In the forest watersheds, Li et al. (2018) (in supplementary) showed that conductivity of baseflow and surface runoff did not change over time. In contrast, agriculture practices such as fertilizer application can influence the concentrations of conductivity and hence affect the CMB method accuracy.

④ Mississippi River basin is the large watershed. The basin has sizeable spatial heterogeneity of climate. The role of climate on hydrology, particularly for low flows are more pronounced in the larger watersheds. It is worth conducting an analysis of this topic. For simplicity, Climate North America (http://climatena.ca/) can provide climate data for the basin.

In sum, further analysis is, for sure, needed to address the knowledge gaps as mentioned above.

Li, Q., Wei, X., Zhang, M., Liu, W., Giles-Hansen, K. and Wang, Y., 2018. The cumulative effects of forest disturbance and climate variability on streamflow components in a large forest-dominated watershed. Journal of Hydrology, 557, pp.448-459.

(3) One recommendation of this manuscript is that the parameters SCro and SCbf can be determined by the 99th percentile and dynamic 99th percentile methods. I agree with the authors to select the 99th percentile of conductivity. However, there is also a concern related to this recommendation. For the CMB method, the SCbf is often corresponding to the lowest flows with potential time lags (Li et al. 2014; in your manuscript). With the recommendation of using the 99th percentile, it might be a chance that the 99th percentile does not correspond to the lowest flows. Therefore, this should be mentioned in the discussion.

(4) The title can be rephrased as "Key challenges facing the application of the conductivity mass balance method: a case study of the Mississippi River Basin".

(5) Table 1 should be reorganized. It is meaningless to use the site number. I suggest the site characteristic such as watershed area, relief, slope, and climate, can also be listed in table 1. As such, the sensitivity can be compared with watershed characteristics.

(6) The objective should be concise. In Line 85, "to resolve some of the questions". Please be more specific, which questions you are going to resolve in this manuscript.

(7) Section 2.5, the meaning of the sensitivity and uncertainty should be elaborated more. For instance, larger values of sensitivity indicate higher sensitivity. A similar explanation is needed for uncertainty.

(8) The language should be polished by the professionals before publication. Here I list some of the suggestions while I read the manuscript.

First two sentences in the abstract. Suggestion: The conductivity mass balance (CMB) method has a long history of application to baseflow separation studies, which uses site-specific and widely available discharge and specific conductance data.

Line 17, insert "in" ; the parameter in the method

Lines 45-47, rewrite

Line 125, the key parameters need to be calculated

Line 140, for at least 5 years

Line 147, delete unbroken

Figure 8 is not clear. Please redraw.

**2. Author's response to the referees and corresponding changes in the manuscript**

**Answer to the comment of Referee #1**

We would like to thank Anonymous Referee #1 for reading our manuscript and for his careful and useful review. Here are our answers to the points raised by his remarks.

**1. In Line133, page 5, it has mentioned that "assigning the 99th percentile (ordered by increasing conductivity) of the stream conductivity monitoring record to avoid the impacts of extremely high SCBF estimates on the separation results", please indicate which conditions can cause extremely parameter values?**

**Author's response:**

The main reason of extremely parameter values are human activities. Human activities can significantly affect stream discharge and water quality, thereby disrupting their natural relationship and causing extreme parameter values, in most cases, there will be a maximum. For example, some monitoring sites located adjacent to reservoirs contribute significantly to increased evaporation and higher conductivity, others located in urban areas may be affected by urban non-point pollution, including irrigation, mining activity, the use of salts as deicing agents for roads and so on, which significantly increase the composition of groundwater, showing relatively poor inverse correlations between stream conductivity and discharge.

These conditions have been mentioned in the manuscript, please see **Line134-136, page 5.**

**2. The study has applied both the uncertainty estimation methods of BFI proposed by Yang et al. (2019) and Genereux and Hooper (1998) to determine the parameters and the shortest time series in the present study. Why do we use both methods at the same time and what are the differences between them?**

**Author's response:**

The reasons that we use both methods are as follows: firstly, both the methods can be applied to calculate the uncertainties of BFI, the Genereux and Hooper(1998)

method is a widely used uncertainty estimating equation, and the recent study of Yang et al.(2019) shows that for time series longer than 365 days, random measurement errors in yk or SCk will cancel each other out, and their influence on BFI can be neglected, considering the mutual offset, the uncertainty in BFI would be halved. So the method should be more accurate when the time series longer than 365 days, but it is not applicable when sampling periods are shorter than 12 months. In our study, different time series (longer or shorter than 365 days) of monitoring data need to be analyzed, so both the methods proposed by Yang et al. (2019) and Genereux (1998) are used at the same time to determine the parameters by different time series.

The reason that we use both methods at the same time and what are the differences between them have been added in **Line185-186 and Line191-193, page 7**.

**3. In table1, why not compare the uncertainty results of the various WSCRO determination methods?**

**Author's response:**

The sensitivity analysis results of our study showed that the sensitivity index for SCBF was generally greater than that for SCRO, so more attention has been focused on SCBF to reduce uncertainty in BFI. Typically, several values of SCBF have been determined by yearly dynamic maximum and 99th percentile methods. However, SCRO is only estimated using the minimum or 99th percentile (ordered by decreasing conductivity) method. WSCBF and WSCRO differs in the calculation of standard deviation. WSCBF is the standard deviation of the SCBF multiplied by the t-value ($\alpha$ =0.05; two-tail) from the Student's distribution, while WSCRO is the standard deviation of the lowest 1% of measured SC concentrations multiplied by the t-value ($\alpha$ =0.05; two-tail), causing that various standard deviations can't be calculated and various WSCRO can't be compared.

The difference of calculation methods between WSCBF and WSCRO have been added in **Line175-181, page 6-7**, and the reason that why not compare the uncertainty results of the various WSCRO determination methods has also been added in **Line 232-235, page 9**.

**4. Fig. 1,3,6,7 should be replaced by more clearer pictures.**

**Author's response:**

By modifying settings, sharp images can be showed clearly now.

**5. In the conclusion part, it is suggested that large watersheds in other latitudes and climates should be considered in the future research, so as to compare and verify the conclusions of this study, and to obtain more general guiding methods.**

**Author's response:**

Thank you for your advice. To verify the conclusions, our future studies would be carried out in other large watersheds with different climates, topography and latitudes, maybe in the Australia.

This has been mentioned in the end of the manuscript (**Line456-457, page 20**)

**6. In the future research, it is suggested that the results of this method can be used to identify the parameters of other methods to improve the accuracy of separation results of other methods.**

**Author's response:**

Thank you for your advice. Identifying the parameters of other methods using CMB method can balance the accuracy and speed, some researches have also mentioned this (Stewart et al., 2007; Zhang et al., 2013; Lott and Stewart, 2013). For example, "the RDF (recursive digital filter) method only requires the stream discharge data as input and, therefore, is one of the most readily available methods for baseflow separation in longterm studies. However, the parameters for the RDF method are often subjectively determined, resulting in high uncertainties in the baseflow separation estimations. On the other hand, the CMB method is considered to be more objective because it is based on the direct measurements of streamflow conductivity. However, the data required for the CMB method may not be available for long periods. A linkage between the RDF and the CMB methods can be established by using the baseflow data estimated with the CMB method to calibrate parameters for the RDF model. The calibrated RDF model can then be used for baseflow separation over a longer period

when only discharge data are available (Zhang et al., 2013). So this will also be the main research object in the future.

This has been mentioned in the end of the manuscript (**Line457-458, page 20).**

**7. Reference format is not consistent. It should follow the guidelines of the Journal.**

**Author's response:**

Following the guidelines of the Journal, reference format has been corrected.

**Answer to the comment of Referee #2**

We highly appreciate Anonymous Referee 2 for extensive and generous comments on the manuscript and his/her generally positive impression of our work. Here we briefly respond to the points raised by his remarks.

**1. In this study, more than 200 sites were included in the data analysis. However, some conclusions were drawn from the simple examination, which lacks the robust evidence of whether these conclusions will hold. For instance, A) the impacts of topography and altitude (Line 276) is concluded by a simple spatial plot (figure 7). In my view, such suggestion is acceptable, but not robust. I suggest the authors can make a scatter plot the correlation against the median elevation of sub-watersheds or other indices that can represent the watershed topography. B)Impact of anthropogenic factors. In this section, the authors only discussed the reservoir as an indicator of human interruption. Disapprovingly, authors only mention the evaporation. The reservoir/dam can provide substantial sources of water in the low flow periods. This may decrease the conductivity in streams and hence undermines the groundwater contribution to streams and leads to an underestimate of baseflow conductivity. Besides, there are other anthro- pogenic factors such as groundwater pumping and agriculture activities that affect the**

**conductivity in streams and should be discussed in the manuscript.**

**Author's response:**

Thank you for your advice. We have made a scatter plot of the correlation against the elevation of sites (Fig.8), it can be found that with the increase of site elevation, the proportion of sites with a correlation coefficient less than -0.5 increased significantly, and it's consistent with our previous conclusions (most stations located in stream headwater areas, with an elevation above 1,500 m, a steep terrain and high elevation showed inverse correlations between flow and conductivity). However, the relationship between them does not strictly satisfy the inverse correlation, and we also found that there are also many sites below 1500 meters (especially 500 meters) that meet the requirements of the correlation coefficient (less than -0.5), these sites mainly located in the Ohio River Basin, the terrain of the basin is relatively flat and the altitude is low, the elevation of many sites located in stream headwater areas are still less than 500 meters, so the impact of elevation in this sub-watershed is not significant. These further analyses have been added to the revised manuscript **(Fig.8 and Line298-305, page 14)**.

Impact of anthropogenic factors. We researched the relevant literature again. As the Referee mentioned, "there are other anthropogenic factors that affect the conductivity in streams and should be discussed in the manuscript." We found that changes in river conductivity can have many different causes, and the major impact processes include agriculture practices (such as fertilizer application), mining activity, the use of salts as deicing agents for roads (Miguel et al., 2013; Crosa et al., 2006; Palmer et al., 2010; Bäthe and Coring, 2011; Dikio, 2010; Kaushal et al., 2005). Besides, other anthropogenic factors such as discharge from industrial activities (Piscart et al., 2005b; Dikio, 2010), sewage treatment plant effluents (Silva et al., 2000; Williams et al., 2003; Lerotholi et al., 2004) or reduced river discharge due to damming (Mirza, 1998) can also cause the variation of conductivity. Admittedly, previous understanding of the impact of anthropogenic factors is one-sided. The influence of reservoir/dam will be revised and other factors have been discussed in the manuscript (**Line340-373, page 16-17, and the following references**).

Miguel Cañedo-Argüelles, Ben J. Kefford, Christophe Piscart, Narcís Prat, Ralf B. Schäfer, Claus-Jürgen Schulz. Salinisation of rivers: An urgent ecological issue, Environmental Pollution 173 (2013) 157-167.

Crosa, G., Froebrich, J., Nikolayenko, V., Stefani, F., Galli, P., Calamari, D., 2006. Spatial and seasonal variations in the water quality of the Amu Darya River (Central Asia). Water Research 40 (11), 2237-2245.

Palmer, M.A., Bernhardt, E.S., Schlesinger, W.H., Eshleman, K.N., Foufoula- Georgiou, E., Hendryx, M.S., Lemly, A.D., Likens, G.E., Loucks, O.L., Power, M.E., White, P.S., Wilcock, P.R., 2010. Mountaintop mining consequences. Science 327 (5962), 148-149.

Bäthe, J., Coring, E., 2011. Biological effects of anthropogenic salt-load on the aquatic fauna: a synthesis of 17 years of biological survey on the rivers Werra and Weser. Limnologica 41(2), 125-133.

Dikio, E.D., 2010. Water quality evaluation of Vaal river, Sharpeville and Bedworth lakes in the Vaal region of south Africa. Research Journal of Applied Sciences, Engineering and Technology 2 (6), 574-579.

Kaushal, S.S., Groffman, P.M., Likens, G.E., Belt, K.T., Stack,W.P., Kelly, V.R., Band, L.E., Fisher, G.T., 2005. Increased salinization of fresh water in the northeastern United States. Proceedings of the National Academy of Sciences of the United States of America 102 (38), 13517-13520.

Piscart, C., Moreteau, J.-C., Beisel, J.-N., 2005. Biodiversity and structure of macroinvertebrate communities along a small permanent salinity gradient (Meurthe river, France). Hydrobiologia 551 (1), 227-236.

Silva, E.I.L., Shimizu, A., Matsunami, H., 2000. Salt pollution in a Japanese stream and its effects on water chemistry and epilithic algal chlorophyll-a. Hydrobiologia 437 (1), 139-148.

Williams, M.L., Palmer, C.G., Gordon, A.K., 2003. Riverine macroinvertebrate responses to chlorine and chlorinated sewage effluents e acute chlorine tolerances of Baetis harrisoni (Ephemeroptera) from two rivers in KwaZulu-Natal, South Africa. Water SA 29 (4), 483-487.

Lerotholi, S., Palmer, C.G., Rowntree, K., 2004. Bioassessment of a River in a Semiarid, Agricultural Catchment, Eastern Cape. In: Proceedings of the 2004 Water Institute of Southern Africa (WISA) Biennial Conference, Cape Town, South Africa, pp. 338-344.

Mirza, M.M.Q., 1998. Diversion of the Ganges water at Farakka and its effects on salinity in Bangladesh. Environmental Management 22 (5), 711-722.

**2. The authors stressed that there is a large amount of watershed where CMB can not be applied. The following question is why this happens in this watershed? I assume that further tests are needed to answer this question. Based on my experience, I suggest the authors can test, but not limited to, the following variables: A) watersheds area, B) Watershed locations, C) snow and rain dominated hydrological regimes, D) Land cover and land use. E) Climate regions.**

**2.1 In Figure 8, authors only examined the relationship between correlation and watershed area in two sub-watersheds. Why don't you examine such relationship for all study watersheds? In smaller watersheds, low flows are mainly fed by groundwater. In contrast, there is always a large amount of surface runoff in the low flows period due to the spatial heterogeneity of climate. In my opinion, you could test all sub-watersheds as well as the entire Mississippi River Basin, and it can drive a threshold of watershed area, above which the CMB methods cannot be applied.**

**2.2 In this study, the authors concluded that headwater watersheds have a better rela-tionship between discharge and conductivity. I assume this is likely due to differences in hydrological regimes, i.e., snow-dominated and rain-dominated. In upper streams, high flows are mainly stimulated by the snow-melt process (e.g., Dyer, 2008). They can be classified as snow-dominated watersheds, while lower watersheds are more likely to be rain-dominated systems. Two systems have a distinct hydrological process, and there is potential uncertainty whether there is a significant difference between the two systems.**

**Dyer, J., 2008. Snow depth and streamflow relationships in large North**

**American watersheds. Journal of Geophysical Research: Atmospheres, 113(D18).**

**2.3 Land cover and land use can be a factor. Forest cover and agriculture land use can have different conductivity concentrations. In the forest watersheds, Li et al. (2018) (in supplementary) showed that conductivity of baseflow and surface runoff did not change over time. In contrast, agriculture practices such as fertilizer application can influence the concentrations of conductivity and hence affect the CMB method accuracy.**

**2.4 Mississippi River basin is the large watershed. The basin has sizeable spatial heterogeneity of climate. The role of climate on hydrology, particularly for low flows are more pronounced in the larger watersheds. It is worth conducting an analysis of this topic. For simplicity, Climate North America (http://climatena.ca/) can provide climate data for the basin.**

**In sum, further analysis is, for sure, needed to address the knowledge gaps as mentioned above.**

**Li, Q., Wei, X., Zhang, M., Liu, W., Giles-Hansen, K. and Wang, Y., 2018. The cumulative effects of forest disturbance and climate variability on streamflow components in a large forest-dominated watershed. Journal of Hydrology, 557, pp.448-459.**

**Author's response:**

Thanks for your advice and experience. The suggestions and questions you have proposed are very good research topics, which are worthy of our special research from the aspects you mentioned in the future study.

**2.1 Watershed area**. We did try to obtain a threshold of watershed area aiming at all sub-watersheds, but the results are not ideal. We found that the watershed areas differ widely among different sub-watersheds. For example, all stations in the Missouri River Basin drain catchments with areas of < 9,000 km$^2$, whereas many catchments exceed 30,000 km$^2$ in the Arkansas River Basin. It can't drive a threshold of watershed area, above which the CMB methods cannot be applied aiming at all sub-watersheds. Therefore, only two sub-watersheds are listed and discussed in the manuscript. And to

avoid confusion, this has been explained in the revised manuscript(**Line325-328, page 14**).

**2.2 Hydrological or hydrogeological regimes? Or both of them?**

According to the distribution of correlation coefficient, we have made the conclusion that headwater watersheds have a better relationship between discharge and conductivity. As you mentioned, "this is likely due to differences in hydrological regimes, i.e., snow-dominated and rain-dominated". To solve this problem, we have further consulted relevant literature. A similar study in Upper Colorado River Basin suggest that there is typically greater baseflow yield in higher elevation watersheds (Rumsey et al., 2015), and Dyer(2008) found that in upper streams, high flows are mainly stimulated by the snow-melt process. But from these findings which are based on the major river basins in North America we still can't establish a relationship between hydrological regimes and the applicability of CMB method (quantitatively expressed by correlation coefficient between discharge and conductivity in our study). We still have the opinion that hydrogeological conditions are more dominant, there is a strong hydraulic connection between groundwater and surface water due to the erosion in upper streams under natural condition, and that the major direction of surface water-groundwater interaction is from groundwater to surface water. In this way, conductivity and streamflow data can accurately reflect the natural spatial and temporal variation. The differences in hydrological regimes, i.e., snow-dominated and rain-dominated will be discussed in our future study, and the above discussion has been added in the manuscript(**Line384-392, page 17-18**).

Christine A. Rumsey, Matthew P. Miller, David D. Susong, Fred D. Tillman, David W. Anning., Regional scale estimates of baseflow and factors influencing baseflow in the Upper Colorado River Basin. Journal of Hydrology: Regional Studies, 4 (2015) 91–107.

Dyer, J., 2008. Snow depth and streamflow relationships in large North American watersheds. Journal of Geophysical Research: Atmospheres, 113(D18).

**2.3 Land cover and land use factor.**

Thank you for your advice, and we are going to attribute the impact of different

land cover and land use to anthropogenic factors, we think that CMB method maintains relatively high accuracy naturally, where less impact of anthropogenic activities happen there, such as forest cover land. In contrast, CMB method is relatively poorly applied to agriculture land with more human intervention to the hydrological process. This has been discussed in the revised manuscript (**Line343-346, page 16**).

**2.4 The role of climate.**

Thank you for your advice, as you mentioned, the role of climate on baseflow are pronounced in the larger watersheds, which is a topic worthy of conducting a special research. We produced a superimposed map of climate zoning and the applicability of CMB method. However, from this figure, it is difficult to summarize the influence rule of climate type on the applicability of CMB method. According to the website provided by the reviewer, we can further obtain meteorological data, and detailed analyze the influence of meteorological factors on the base flow in future studies. This has been mentioned in the revised manuscript (**Line386-389, page 17**).

[Figure]

**Climate type and spatial distribution of correlation coefficients for the correlation between stream conductivity and discharge in the Mississippi River Basin**

In sum, in order to confirm the reasons that some watershed where CMB can't be applied depends on an overall study in the future, thanks again to the anonymous reviewers for their suggestions.

**3. One recommendation of this manuscript is that the parameters SCro and SCbf can be determined by the 99th percentile and dynamic 99th percentile methods. I agree with the authors to select the 99th percentile of conductivity. However, there is also a concern related to this recommendation. For the CMB method, the SCbf is often corresponding to the lowest flows with potential time lags (Li et al. 2014; in your manuscript). With the recommendation of using the 99th percentile, it might be a chance that the 99th percentile does not correspond to the lowest flows. Therefore, this should be mentioned in the discussion.**

**Author's response:**

Thank you for asking the question that makes our research more rigorous. We have checked the parameters determined by dynamic 99th percentile method and didn't found the condition that the 99th percentile doesn't correspond to the lowest flows. However, it must be stressed that although the applicability of the CMB method has been verified for a site before determining parameters, it cannot be guaranteed that there will be no anthropogenic disturbance to parameters of a site in which the CMB method has been found to be applicable, and that the parameters correspond to the lowest flows very well. For example, leakage of an underground storage tank may last for a long time, which may result in many observations of extremely high conductivities that cannot be avoided by the 99$^{th}$ percentile method. So there is a possibility that the 99$^{th}$ percentile conductivity does not correspond to lowest flows. Therefore, parameters should assessed after calculation by the 99$^{th}$ percentile method to further avoid abnormal phenomena and errors within separation results. Based on the above analysis, we have suggested in our revised manuscript that parameters should be checked after calculating by 99th percentile method to further avoid abnormal phenomena (**Line410-416, page 18**).

**4. The title can be rephrased as "Key challenges facing the application of the**

**conductivity mass balance method: a case study of the Mississippi River Basin"**.

**Author's response:**

Thank you for your advice and it has been rephrased in our revised manuscript(**Line 1-2, page 1**).

**5. Table 1 should be reorganized. It is meaningless to use the site number. I suggest the site characteristic such as watershed area, relief, slope, and climate, can also be listed in table 1. As such, the sensitivity can be compared with watershed characteristics.**

**Author's response:**

Thank you for your advice and the Table has been reorganized in our revised manuscript. Site characteristics such as watershed area, elevation and slope have been listed in table 1 to be compared with sensitivity.

**6. The objective should be concise. In Line 85, "to resolve some of the questions". Please be more specific, which questions you are going to resolve in this manuscript.**

**Author's response:**

The sentence has been modified to be more concise in the revised manuscript(**Line 85-87, page 3**).

**7. Section 2.5, the meaning of the sensitivity and uncertainty should be elaborated more. For instance, larger values of sensitivity indicate higher sensitivity. A similar explanation is needed for uncertainty.**

**Author's response:**

Thank you for your advice and the relative explanation will be added in our revised manuscript.

Previous manuscript (**Line 220-225, page 9**) expound the meaning of the sensitivity, to make it clearer, a sentence has been added to the revised manuscript (**Line 168, Line 173-175, page 6**)

**8. The language should be polished by the professionals before publication. Here I list some of the suggestions while I read the manuscript.**

**First two sentences in the abstract. Suggestion: The conductivity mass balance (CMB) method has a long history of application to baseflow separation studies, which uses site-specific and widely available discharge and specific conductance data.**

**Line 17, insert "in" ; the parameter in the method**

**Lines 45-47, rewrite**

**Line 125, the key parameters need to be calculated**

**Line 140, for at least 5 years**

**Line 147, delete unbroken**

**Figure 8 is not clear. Please redraw.**

**Author's response:**

Thank you for your correction, all of them has been revised in the manuscript(**Line 8-10, page 1; Line 16, page 1; Lines 45-48, rewrite; Line 124, page 5; Line 140, page 5; Line 147, page 5; Figure 9**), and the language has been polished by the professionals before publication(EDITORIAL CERTIFICATE)

[Figure]

https://www.mjeditor.com

**EDITORIAL CERTIFICATE**

The English writing of the following manuscript was carefully edited by a native English speaker.

**Manuscript information**

ID:MJ2020010098699

Editing date:2020.10.09

Title:Key challenges facing the application of the conductivity mass balance method: a case study of the Mississippi River Basin.

Author:Hang Lyu, Chenxi Xia, Jinghan Zhang, Bo Li.

Language writing before editing: ☐Very poor ☐Poor ☑Fair ☐Good ☐Very good ☐Excellent

Recommendation after language editing
☑Submitting to target journal directly
☐Submitting to target journal after minor revision
☐Re-editing required after major revision
☐Not suitable for publication

**Certificate by**

*Saphiya.K*

Editor in Chief
MJ Language Editing Services, Shenzhen, China

MJ Language Editing Services, offers professional English language editing and publication support services to authors engaged in over 500 areas of research through its community of experienced editors, which includes doctors, published scientists, and researchers with peer review experience. Authors who work with MJ are guaranteed excellent language quality and timely delivery.

**MJ Language Editing Services**

Diwang Building, No. 5002 Shennan Road, Luohu District, Shenzhen, China

Tel:+086 0755 25100506

**Marked-up manuscript.**

[revised manuscript text omitted]

---

## Referee Report (RR1)

I am reviewer #2 in the first round of review for the manuscript of "key challenges facing the application of the conductivity mass-balance (CMB) method: a case study of the Mississippi River Basin". The authors have addressed all reviewers' comments and made a substantial improvement in the manuscript. However, I still have some minor suggestions, which will make this manuscript even stronger for this research theme. Therefore, I suggest a minor revision for this revised manuscript. Also, in my view, no further review is needed when the authors submit their revision only if the comments are carefully revised and figures in the manuscript are modified.

1. Sensitivity analysis for the conductivities of baseflow and surface runoff

In the manuscript, the takeaway messages from the sensitivity analysis I can get are that the baseflow index is more sensitive to the conductivity of baseflow sensitivity (BFsc) than the conductivity of surface runoff (SRsc). This conclusion has been mentioned in several case studies. The contribution of your study is to explain the uncertainty of BFsc and SRsc in the uncertainty of the baseflow index. However, when someone adopts the CMB method for baseflow separation, the concerns are how big the BFI errors are when BFsc and SRsc are over- or under-estimated by a certain percentage. For instance, in the sensitive analysis of Zhang et al. (2013), (which you labelled as 2012 in your paper, correct it for your next revision), if BFsc had been underestimated by 20%, BFI would have been overestimated by 26%. Overestimation of BFsc, however, would have less impact on BFI compared with the underestimation of the same parameter. In my view, your sensitivity analysis is good but still needs a step forward.

2. Groundwater pumping impacts on baseflow and conductivity should be discussed.

The authors did a great job in the revised version to provide sufficient discussion on the human impacts on baseflow. A short paragraph of the groundwater pumping impacts on baseflow and conductivity are needed. For instance, groundwater pumping can reduce groundwater discharge to stream and/or induce stream infiltration to the aquifer, leading to streamflow depletion (Gleeson and Ritcher, 2018).

Gleeson, T., & Richter, B. (2018). How much groundwater can we pump and protect environmental flows through time? Presumptive standards for conjunctive management of aquifers and rivers. River research and applications, 34(1), 83-92.

3. Editorial changes in the revised manuscript.
   a. Figure 1
Can you show the location in North American or the USA for the readers to locate the watershed as HESS is an international journal?
   b. 99th percentile
In your manuscript, you used different standards for ranking, i.e., increasing and decreasing. I understand that this is a more consistent expression and also related to the physical meaning of baseflow and surface runoff conductivity. However, this will also cause confusion as readers may think you are using the same ranking order. I suggest you use the 99th and 1st percentile.
   c. Figure 3,
Can you make your points bigger?
   d. Figure 6

Do not fill the boxes in figures as this overlaps the streamflow data.

        e.   Figure 7

No legend is provided in the figure.

Overall, thanks for your revision and nice work!

---

## Author Response (AR2)

**Response to the reviews on "Discussion on key challenges facing the application of the conductivity mass-balance (CMB) method: a case study of the Mississippi River Basin" by Hang Lyu et al.**

We highly appreciate again for extensive and generous comments on the manuscript and his/her generally positive impression of our work. Here we briefly respond to the points raised by his remarks.

1. Sensitivity analysis for the conductivities of baseflow and surface runoff In the manuscript, the takeaway messages from the sensitivity analysis I can get are that the baseflow index is more sensitive to the conductivity of baseflow sensitivity (BFsc) than the conductivity of surface runoff (SRsc). This conclusion has been mentioned in several case studies. The contribution of your study is to explain the uncertainty of BFsc and SRsc in the uncertainty of the baseflow index. However, when someone adopts the CMB method for baseflow separation, the concerns are how big the BFI errors are when BFsc and SRsc are over- or under-estimated by a certain percentage. For instance, in the sensitive analysis of Zhang et al. (2013), (which you labelled as 2012 in your paper, correct it for your next revision), if BFsc had been underestimated by 20%, BFI would have been overestimated by 26%. Overestimation of BFsc, however, would have less impact on BFI compared with the underestimation of the same parameter. In my view, your sensitivity analysis is good but still needs a step forward.

**Author's response:**

   Thank you for your advice. We have conducted a supplementary analysis to investigate that how big the BFI errors are when BFsc and SRsc are over- or under-estimated by a 10%. And it can be proved that although underestimation or overestimation of $SC_{BF}$ of the same degree, the former one has more impact on BFI. This has been discussed in the revised manuscript (Line 222-228, Page 9 and Line 406-407, Page 18). On the other hand, the cited literature of Zhang et al. has been correct to be 2013.

2. Groundwater pumping impacts on baseflow and conductivity should be discussed. The authors did a great job in the revised version to provide sufficient discussion on the human impacts on baseflow. A short paragraph of the groundwater pumping impacts on baseflow and conductivity are needed. For instance, groundwater pumping can reduce groundwater discharge to stream and/or induce stream infiltration to the aquifer, leading to streamflow depletion (Gleeson and Ritcher, 2018).

**Author's response:**

Thanks for your advice. Groundwater pumping can reduce groundwater discharge to stream and affect the hydraulic connection between groundwater and surface water, then invalidates the application of the CMB method. When a well is pumped at a constant rate, initially most of the groundwater comes from storage, eventually reaching the river, inducing a leakage of stream water to adjacent aquifer and depleting streamflow significantly (Bredehoeft and Kendy, 2008; Gleeson and Ritcher, 2018). This change in relationship between groundwater and surface water renders CMB method less applicable. These have been discussed in the revised manuscript (Line 370-374, Page 17 and Line 342, Page 16).

3. Editorial changes in the revised manuscript.
   a. Figure 1
   Can you show the location in North American or the USA for the readers to locate the watershed as HESS is an international journal?
   b. 99th percentile
   In your manuscript, you used different standards for ranking, i.e., increasing and decreasing. I understand that this is a more consistent expression and also related to the physical meaning of baseflow and surface runoff conductivity. However, this will also cause confusion as readers may think you are using the same ranking order. I suggest you use the 99th and 1st percentile.
   c. Figure 3,
   Can you make your points bigger?
   d. Figure 6
   Do not fill the boxes in figures as this overlaps the streamflow data.
   e. Figure 7
   No legend is provided in the figure.

Thanks for the Editorial changes. The Figure 1,3,6,7 have been modified, please see the corresponding picture in the article, and the method name of determining the parameter $SC_{RO}$ has been changed to 1st percentile method.

---

## Author Response (AR3)

**Response to the reviews on "Discussion on key challenges facing the application of the conductivity mass-balance (CMB) method: a case study of the Mississippi River Basin" by Hang Lyu et al.**

We highly appreciate again for comments on the manuscript. Here we briefly respond to the points raised by the editor.

1. In my final review of the manuscript, I noticed that some of the tables and figures (Table 1, Figure 6, Figure 8 (no units on the x-axis) and Figure 10) do not express the units in SI units, as required by the HESS guidelines (https://www.hydrology-and-earth-system-sciences.net/submission.html#math). I am very sorry that I did not catch this sooner. Once these changes are made, the manuscript is ready for publication in HESS.

**Author's response:**

Thank you, besides Figure 6, 8, 10 and Table 1, Figure 2, 9 have also been modified, all the units have been changed, please see the corresponding picture in the article.